# MesaTask: Towards Task-Driven Tabletop Scene Generation via 3D Spatial Reasoning

**Jinkun Hao**[1][*] **Naifu Liang**[2][*], **Zhen Luo**[3,4][*], **Xudong Xu**[2][‡], **Weipeng Zhong**[2], **Ran Yi**[1][†],
**Yichen Jin**[5], **Zhaoyang Lyu**[2], **Feng Zheng**[4], **Lizhuang Ma**[1][†] **Jiangmiao Pang**[2]

[1]Shanghai Jiao Tong University    [2]Shanghai AI Laboratory    [3] SII
[4]Southern University of Science and Technology    [5]Peking University

Project page: https://mesatask.github.io/

## Abstract

The ability of robots to interpret human instructions and execute manipulation tasks necessitates the availability of task-relevant tabletop scenes for training. However, traditional methods for creating these scenes rely on time-consuming manual layout design or purely randomized layouts, which are limited in terms of plausibility or alignment with the tasks. In this paper, we formulate a novel task, namely task-oriented tabletop scene generation, which poses significant challenges due to the substantial gap between high-level task instructions and the tabletop scenes. To support research on such a challenging task, we introduce **MesaTask-10K**, a large-scale dataset comprising approximately 10,700 synthetic tabletop scenes with *manually crafted layouts* that ensure realistic layouts and intricate inter-object relations. To bridge the gap between tasks and scenes, we propose a **Spatial Reasoning Chain** that decomposes the generation process into object inference, spatial interrelation reasoning, and scene graph construction for the final 3D layout. We present **MesaTask**, an LLM-based framework that utilizes this reasoning chain and is further enhanced with DPO algorithms to generate physically plausible tabletop scenes that align well with given task descriptions. Exhaustive experiments demonstrate the superior performance of MesaTask compared to baselines in generating task-conforming tabletop scenes with realistic layouts.

## 1 Introduction

A fundamental challenge in robotic manipulation is enabling robots to accurately interpret human instructions and successfully execute complex tasks accordingly. The conventional pipeline for achieving this involves task definition, simulatable tabletop scene construction, and policy training. However, traditional scene construction methods, which rely on manual design or purely randomized layouts, are limited by their labor-intensive nature and the resulting constraints on diversity and plausibility, ultimately hindering the generalization of learned policies. Therefore, automatic *task-oriented* tabletop scene generation emerges as a promising approach for effectively bridging the gap between task descriptions and scenes. Crucially, tabletop scene generation must satisfy three key requirements: covering task variables, enabling scene interactivity, and ensuring realistic layouts, thereby facilitating the learning of robust policies.

Existing scene generation methods [27, 6, 11] often start from a single scene image and attempt to recover the corresponding tabletop scene through object retrieval and layout optimization. Unfortunately, their ability to understand under-specified task instructions still requires empirical corroboration. Other approaches [37, 35, 2] leverage powerful language models (LLMs) to interpret

---

[*]Equal contribution, ‡ Project lead
[†]Corresponding authors

39th Conference on Neural Information Processing Systems (NeurIPS 2025).

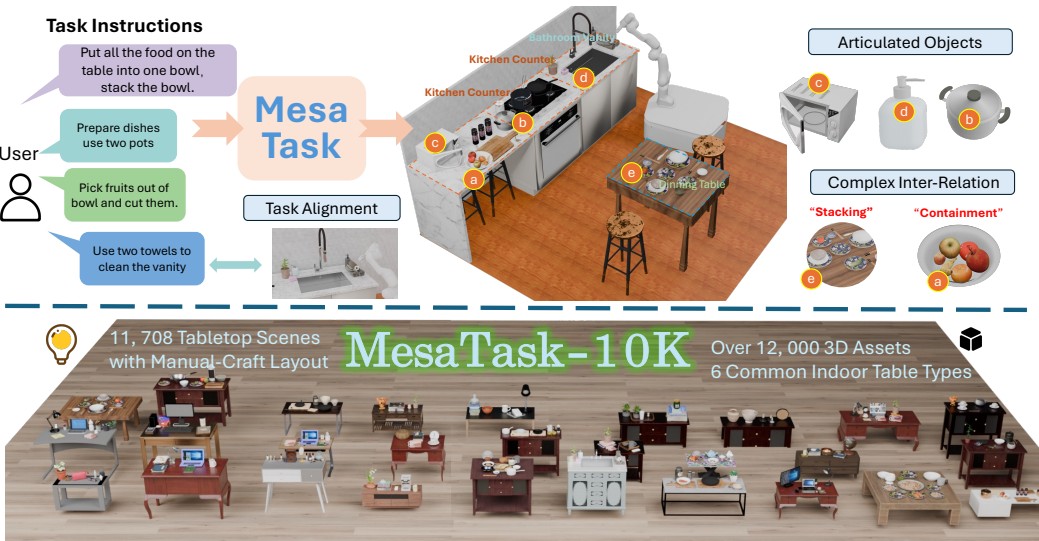

Figure 1: We present MesaTask, a novel LLM-based framework for generating task-oriented 3D tabletop scenes directly from high-level human instructions, featuring realistic layouts, articulated objects, and complex inter-object relations like stacking and containment. To support this task, we introduce a large-scale dataset of tabletop scenes, MesaTask-10K, comprising over 12, 000 3D assets, 11, 708 tabletop scenes with manually crafted layouts covering 6 common indoor table types.

task prompts and then synthesize tabletop scenes in a *zero-shot* manner. Nevertheless, these methods are hindered by inherent limitations, no matter the inevitable occlusion in scene images or the lack of fine-tuning on a scene dataset, which significantly impede the modeling of realistic table layouts and complex inter-object relations, such as stacking and containment, within the scene. As a result, task-oriented tabletop scene generation remains a challenging problem due to the scarcity of datasets and the substantial gap between task instructions and scene layouts.

To tackle these challenges, we collect a first-of-its-kind dataset of synthetic tabletop scenes with *manually crafted layouts*, dubbed **MesaTask-10K**. As shown in Figure 1, our dataset comprises approximately 10, 700 diverse tabletop scenes, spanning six common indoor table categories, including office tables, dining tables, kitchen counters, and more. The 3D objects in MesaTask-10K originate from a large asset library containing over 12, 000 rigid and articulated 3D assets, each with detailed semantic information, such as object category, description, and materials, and featuring a comprehensive taxonomy of over 200 object classes on the tables. As claimed in [27], pretrained 2D image generative models better capture scene and object configurations both at the scene level and in fine-grained inter-object relations. Inspired by this, our dataset is built upon diverse tabletop scene images with diversity and realistic layouts, generated by a large text-to-image model [12] pretrained on massive internet data. To obtain a coarse layout from the scene image, we estimate the depth [33] of each image, extract the instance point cloud, and acquire the 3D bounding box of objects. We then leverage the object descriptions labeled by VLM [1] to retrieve suitable 3D assets from the library and construct an initial replica of the tabletop scenes. Subsequently, human annotators meticulously refine these 3D layouts, adjusting the object size and positions as per the image prompt, addressing inaccuracies from occlusion and ensuring complex inter-object relations. Ultimately, all the scenes are put into a physical simulator, IsaacSim [19], to prevent object collisions.

Confronted with the significant gap between tasks and scenes, we propose a novel paradigm referred to as **Spatial Reasoning Chain**, decomposing task-to-tabletop scene generation into a structured chain of thought (CoT). Given a high-level task description, this chain of thought begins with the inference of requisite objects, accompanied by their semantic attributes and spatial interrelations, based on which a complete scene graph is formed, and finally leads to a concrete 3D layout of objects on the table. To establish trainable spatial reasoning chains with our dataset, we design a set of delicate rules to extract the object attributes and inter-object relations, thus forming a scene graph for each tabletop scene. Subsequently, we leverage a multimodal large language model, taking scene graphs and rendered scene images as input, to generate corresponding task information and detailed spatial reasoning descriptions for training.

Thanks to our structured reasoning chains, it's convenient to empower LLM with 3D spatial reasoning and scene generation capability. In this paper, we propose **MesaTask**, a novel LLM-based framework for task-oriented tabletop scene generation. Specifically, we initially employ the supervised fine-tuning (SFT) strategy on our constructed reasoning data to inject the LLM with 3D spatial reasoning capabilities. However, MesaTask occasionally generates unsatisfactory tabletop scenes with minor object collisions and misalignment with the given task. To circumvent this hurdle, we devise paired training data and leverage a conventional RL algorithm, namely Direct Preference Optimization (DPO), to boost our MesaTask model, thereby ensuring that the generated scenes are devoid of object collisions and exhibit improved conformity with the provided task descriptions.

For a more comprehensive performance assessment, we leverage powerful VLMs to evaluate the rendered scene images from multiple perspectives, including task-scene alignment, physical viability, scene layout plausibility, *etc*. Through extensive experiments, our MesaTask framework is capable of generating physically plausible tabletop scenes with realistic layouts, outperforming baseline methods in terms of FID, VLM-based metrics, and the user study. In particular, our generated tabletop scenes strictly conform to given task instructions and exhibit rich inter-object relations, such as stacking or containing. In summary, our contributions are threefold:

- We pioneer the formulation of *Task-to-Scene* generation task, which aims to generate physically plausible tabletop scenes directly from high-level task descriptions.
- We introduce **MesaTask-10k**, a large-scale tabletop scene dataset with human-crafted realistic layouts, characterized by rich inter-object relations and a tremendous amount of synthetic 3D object assets.
- Along with the delicate design of spatial reasoning chains, we propose **MesaTask**, an LLM-based framework endowed with the capability of 3D spatial reasoning and tabletop scene generation, achieving superior performance across various evaluation criteria.

## 2 Related Work

**Tabletop Scenes Dataset.** Recent works have explored various approaches to constructing tabletop scene datasets. LVDiffusor [39] uses large VLMs to generate semantically plausible tabletop scene images, which are constrained in the 2D domain and lack 3D spatial information. StructFormer [17] and StructDiffusion [16] collect 3D object rearrangement data under language guidance using abstract geometric relations, which allows testing structural reasoning but lacks semantic richness and realism for real-world deployment. SetItUp [30] presents manually designed functional scenes reflecting real-world usage like dining or working, but the object sets are fixed and lack diversity. TO-Scene [29] provides a large-scale and richly annotated 3D tabletop dataset built by professional designers. However, its top-down, click-to-place annotation paradigm restricts the inclusion of intricate spatial relationships such as nesting and stacking. Despite these efforts, existing datasets frequently exhibit limitations in terms of data scale, layout, or realism. Accordingly, we introduce a large-scale tabletop scene dataset with diverse real-world 3D objects, realistic layouts, and rich 3D spatial relationships.

**Scene Reconstruction from A Single Image.** It's a long-standing problem to reconstruct 3D scenes from a single image. A line of previous methods [3, 40, 15, 18] attempt to reconstruct the scenes by compressing the input images with an encoder and mapping the image features back to the 3D space via a decoder. Based on advancements in 3D object generation [41, 13], MIDI [11] is capable of generating scenes with diverse 3D objects but struggles to generate complex inter-object relationships. Some other methods [38, 10, 4, 14, 6, 27] typically entail a multi-stage process, comprising object segmentation, occlusion completion, image-to-3D generation, and layout optimization. This protracted workflow inevitably gives rise to error accumulation, particularly in regions with severe occlusions. Moreover, these methods fall short of generating scenes from underspecified task descriptions. In contrast, our LLM-based framework is inherently designed to fit the task-oriented tabletop scene generation.

**LLM-Based Scene Generation.** Inspired by the prosperity of Large Language Models (LLMs), many researchers have exploited the capabilities of powerful LLMs to perform 3D scene generation. For instance, LayoutGPT [8] explores direct 3D layout generation through in-context learning. Furthermore, some methods [9, 2, 37] are built upon commercial LLMs and use multi-stage prompting to achieve open-vocabulary and dataset-free generation in a zero-shot manner. However, these methods encounter substantial challenges in modeling complex inter-object relations. LLPlace [35]

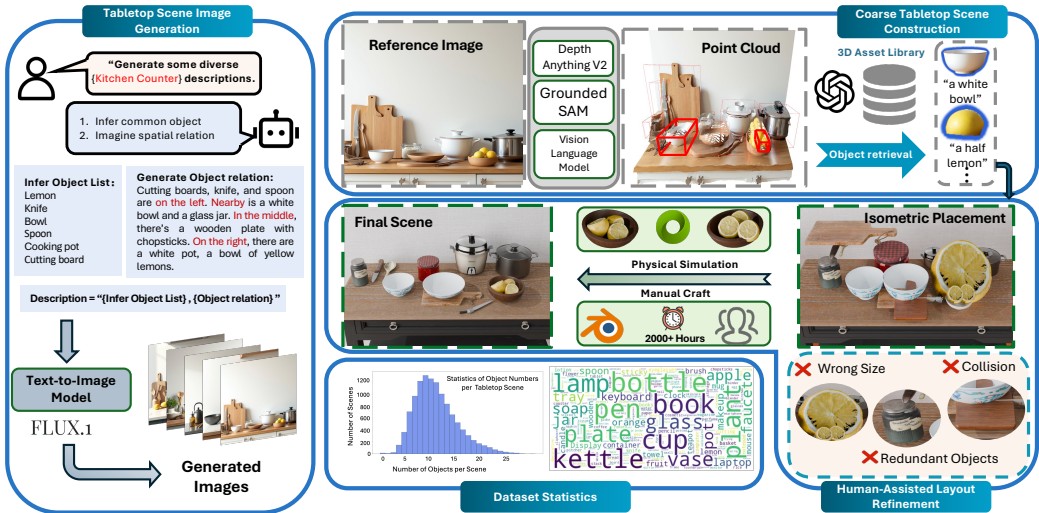

Figure 2: **The dataset construction pipeline.** First, an LLM is used to generate diverse tabletop scene descriptions, including relevant object lists and spatial relations. Conditioned on the scene description, a text-to-image model synthesizes reference images, from which coarse 3D layouts are built using depth estimation, object detection, and 3D asset retrieval. These layouts are refined through human annotations and physical simulation to ensure spatial plausibility, yielding high-quality 3D layouts.

attempts to fine-tune the LLM via supervised fine-tuning (SFT) on meticulously crafted 3D scene datasets, albeit without a specific focus on tabletop scenes. In contrast, we also leverage an LLM-based framework for tabletop scene generation, but our approach involves training on a large-scale scene dataset with manually crafted layouts, thereby empowering our model with superior capabilities for generating realistic layouts and intricate inter-object relationships.

## 3 MesaTask-10K Dataset

Inspired by ARCHITECT [27], we build the MesaTask-10K dataset upon diverse tabletop scene images generated by a pretrained text-to-image model [12], ensuring realistic scene layouts and complex inter-object relationships.

**Tabletop Scene Image Generation**. To better facilitate manipulation tasks, we intend to synthesize diverse tabletop scene images of six common indoor table types in our daily life: office table, dining table, kitchen counter, coffee table, bathroom vanity, and dressing table. As illustrated in Figure 2, the pretrained LLM [1] is guided to output an object list on the table and their spatial relations, respectively, which are subsequently combined to form final scene descriptions. Conditioned on these scene descriptions, FLUX [12], a cutting-edge text-to-image model, is utilized to produce a diverse range of reference scene images *w.r.t.* six distinct table categories.

**Coarse Tabletop Scene Construction.** To create 3D replicas of scene images, we first collect a high-quality 3D asset library through meticulous asset curation from two datasets, namely Obja-verse [7], and PartNet-Mobility [28]. It's noteworthy that our library consists of over $12,000$ rigid and interactive objects along with rich object semantic information, including object category, text descriptions, and materials. As shown in Figure 2, Grounded-SAM [1] is employed to identify all the object instances in the scene image, and a multimodal LLM like GPT-4o is responsible for providing the corresponding semantic information for subsequent 3D object retrieval. During the object retrieval process, we specifically rely on textual descriptions of objects rather than their visual appearance, considering severe occlusions in the tabletop scene images. Furthermore, we utilize Depth Anything v2 [32] to construct the point cloud of tabletop scenes, and leverage instance masks to obtain 3D bounding boxes for each object within the scene, thereby yielding a coarse 3D layout of the tabletop scene.

**Human-Assisted Layout Refinement.** Owing to the intricate inter-object relationships and severe occlusions in the reference images, the obtained coarse scene layouts inevitably contain various flaws, including inaccuracies in object scale, redundant object instances, and object collisions or floating, as shown in Figure 2. To the best of our knowledge, these awkward issues can only be effectively addressed with human assistance. Therefore, 20 expertize annotators undertake a manual layout refinement in Blender, wherein they adjust the object size and positions, as well as eliminate redundant instances, following the reference images.

During annotation, annotators are provided with the coarse 3D scene in GLB format, which includes a Unitree H1 robot model with an absolute height of 1.7m to facilitate the construction of metric-scale 3D scenes, along with reference images and all object snapshots from the images. They will adjust each object's relative size and position with reference to the given tabletop scene images, calibrate the overall scene scale using the H1 model, and rotate objects to match their orientations in the images. On average, annotators spend 10 to 20 minutes on each tabletop scene. Subsequently, we put all tabletop scenes into the physical simulator to prevent object collisions, manually exclude unsatisfactory scenes, and finally create our dataset.

**Dataset Statistics.** MesaTask-10K is a large-scale dataset with approximately $10,700$ diverse tabletop scenes spanning six common indoor table categories. Meanwhile, our curated 3D asset library contains a vast collection of over $12,000$ diverse objects, covering a broad spectrum of more than 200 object classes that are typically encountered on tables. In particular, the 100 object categories that occur the most frequently within this library are visually represented in Figure 2. Moreover, there are roughly 15 objects per tabletop scene on average, and the distribution of the object number is also visualized in Figure 2. We believe that our MesaTask-10K dataset possesses substantial potential to drive research advancements in the realm of task-oriented tabletop scene generation.

## 4 Method

In this section, we present our novel LLM-based framework **MesaTask** for generating realistic 3D tabletop scenes from manipulation task descriptions. The crux of our approach lies in endowing an LLM with the capability of *3D spatial reasoning*, enabling it to infer the complex spatial arrangements necessary to fulfill the requirements of a given task. We formalize the problem in Section 4.1 and describe the system outlined in Figure 3. To empower the LLM with 3D spatial reasoning, we propose a novel spatial reasoning chain in Section 4.2 and introduce our model's training via SFT and DPO algorithms in Section 4.3.

### 4.1 Problem Formulation

Task-oriented tabletop scene generation aims at generating suitable tabletop scenes $\mathbf{S}$ from high-level manipulation task instructions $\mathbf{T}$. Following prior works [34, 25, 35], a tabletop scene $\mathbf{S}$ is a composition of $N$ 3D objects arranged in a specific 3D layout $\mathbf{L} = \{\mathbf{l}_1, \mathbf{l}_2, ..., \mathbf{l}_N\}$. The layout of each 3D object $\mathbf{l}_i = [\mathbf{p}_i, \mathbf{s}_i, \boldsymbol{\theta}_i, \mathbf{t}_i]$ is defined by its location $\mathbf{p} \in \mathcal{R}^3$, axis-aligned 3D bounding box size $\mathbf{s} \in \mathcal{R}^3$, rotation angle around the vertical axis $\boldsymbol{\theta} \in \mathcal{R}$, and its textual description $\mathbf{t}$ detailing the category, shape, and appearance. Given a task instruction $\mathbf{T}$, we will leverage a pretrained LLM model to generate more detailed task information, including the table environment description $\mathbf{E}$, a sequence of decomposed goals $\mathbf{G}$ for this given task, and a set of task-relevant objects $\mathbf{O}$. Based on them, the scene generation model $\mathcal{M}$ is responsible for generating a corresponding 3D scene layout $\mathbf{L}$:

$$\mathbf{L} = \mathcal{M}\left(\mathbf{E}, \mathbf{G}, \mathbf{O}\right), \qquad [\mathbf{E}, \mathbf{G}, \mathbf{O}] = \mathbf{LLM}(\mathbf{T}). \tag{1}$$

Based on the object descriptions in the layout $\mathbf{L}$, appropriate 3D assets will be retrieved from the 3D asset database to form a complete tabletop scene $\mathbf{S}$. It's noteworthy that the generated tabletop scenes will contain all the 3D objects recommended in $\mathbf{O}$ and typically include many other objects to ensure realistic layouts.

### 4.2 Spatial Reasoning Chain

While task instructions are typically conveyed through natural language expressions, 3D scene layouts are inherently represented with structured spatial configurations. Considering the large gap between tasks and tabletop scenes, we propose the spatial reasoning chain to decompose the challenging

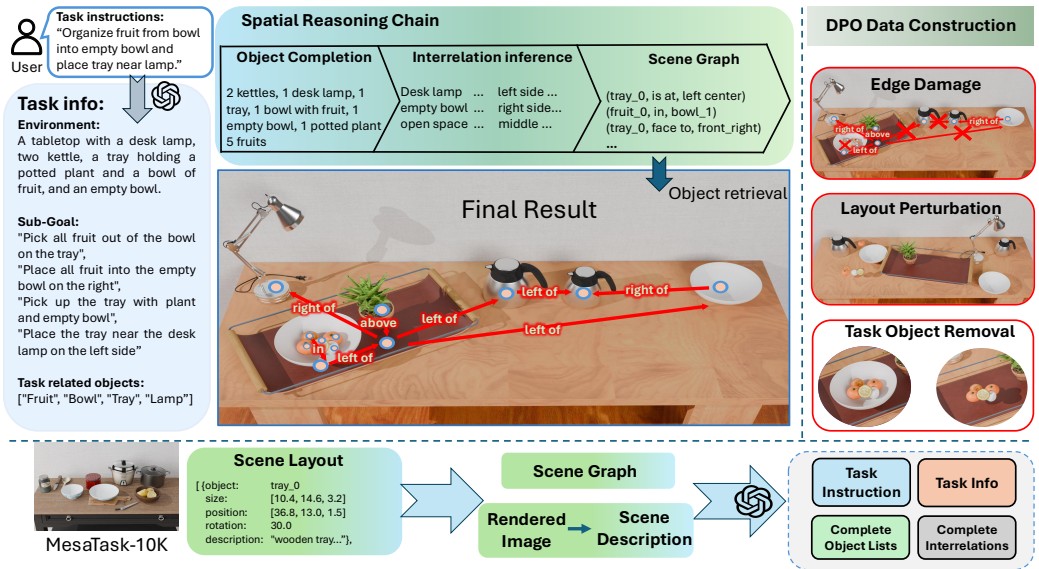

Figure 3: **Overview of our MesaTask Framework. 1)** Task-to-Scene Generation (upper-left). Given a task instruction, we extract detailed task information including environment, sub-goals, and task-relevant objects. A structured spatial reasoning chain performs object list completion, interrelation inference, and scene graph construction, which guides the generation of 3D layouts. Final scenes are obtained via 3D asset retrieval. **2)** Reasoning Data Construction (bottom). Based on scene graphs and descriptions of our MesaTask-10K dataset, A multimodal LLM is leveraged to produce task instructions, detailed task information, and complete object lists and interrelations. **3)** DPO Data Construction (upper right). To enable DPO training, we generate negative examples by randomly perturbing object positions or relations and removing key objects from normal layouts.

task-to-scene generation problem into a structured chain of thought (CoT), significantly easing the training and inference of LLM-based models like ours.

**Task-to-Scene Generation via Spatial Reasoning Chain.** The spatial reasoning chain encompasses three pivotal steps, namely object list completion, interrelationship inference, and scene graph construction, which function as an effective bridge between input tasks and desirable 3D layouts. In the object list completion stage, the generation model $\mathcal{M}$ is motivated to infer a complete list of 3D objects $\mathcal{V}$ given the aforementioned task-relevant objects $\mathbf{O}$. Then, the model $\mathcal{M}$ will generate inter-object relations $\mathcal{E}$ expressed with text descriptions, conditioned on the given task instructions and typical object co-occurrence patterns. With graph nodes $\mathcal{V}$ and graph edges $\mathcal{E}$, the scene graph $\mathcal{G}(\mathcal{V}, \mathcal{E})$ can be represented as:

$$[\mathcal{V}, \mathcal{E}] = \mathcal{M}(\mathbf{E}, \mathbf{G}, \mathbf{O}), \qquad \mathbf{L} = \mathcal{M}(\mathcal{G}(\mathcal{V}, \mathcal{E})) \qquad (2)$$

Notably, to better guide the generation of 3D layouts, we further enrich the graph nodes by incorporating objects' coarse positions and orientations expressed in natural language. Specifically, the orientation is discretized into eight categories, namely front, back, left, right, left-front, left-back, right-front, and right-back, with a 45-degree quantization. The coarse positions correspond to a $3 \times 3$ grid of the table, including center, front, back, left-center, right-center, left-front, right-front, left-back, right-back. Therefore, given a task description, our spatial reasoning chain will prompt the model to sequentially reason about the scene composition, the spatial interrelationship, the scene graph, and ultimately, the 3D scene layout.

**Reasoning Data Construction for Model Training.** To guide the model's reasoning process along our designed spatial reasoning chain, we construct massive reasoning data for training by using our collected tabletop scene dataset, MesaTask-10K. For a certain scene $\mathbf{S}$ inside, we first assign coarse positions and orientations for 3D objects on the table following the quantization rules above, infer the inter-object relations based on the 3D layouts, and finally obtain a complete scene graph $\mathcal{G}_{\mathbf{S}}$. To compensate for the spatial relations missing in the scene graph, we also utilize a multimodal LLM (MLLM) like GPT-4o [1] to output a detailed scene description $\mathbf{D}$ based on a high-quality rendering image $\mathbf{I}$ of the tabletop scene and the scene graph. Given the scene graph $\mathcal{G}_{\mathbf{S}}$ and scene descriptions

**D**, the multimodal LLM is prompted to generate a complete object list $\mathcal{V}$ and inter-object relations $\mathcal{E}$, as well as the corresponding task instructions **T**, in particular including aforementioned detailed task informations z, *i.e.*, [**E**, **G**, **O**] following:

$$\mathbf{D} = \mathbf{MLLM}(\mathbf{I}, \mathcal{G_S}), \quad [\mathbf{T}, \mathbf{E}, \mathbf{G}, \mathbf{O}, \mathcal{V}, \mathcal{E}] = \mathbf{MLLM}(\mathbf{D}, \mathcal{G_S}). \tag{3}$$

### 4.3 LLM-based Framework for Tabletop Scene Generation

Our proposed MesaTask framework is a novel paradigm for tabletop scene generation, comprising an LLM-based model $\mathcal{M}$ for 3D layout generation and a post-processing module responsible for 3D asset retrieval. Given the constructed spatial reasoning data, we perform supervised fine-tuning (SFT) on the MesaTask model, thereby empowering it with the capability to reason about spatial relationships and generate structured 3D layouts from high-level task instructions. Despite the SFT on our high-quality data, the MesaTask model still generates suboptimal 3D layouts, including minor object collisions, unreasonable inter-object relationships misaligned with the task, and the omission of crucial task-relevant objects. Accordingly, we employ the Direct Preference Optimization (DPO) algorithm[21] to tackle such issues.

**DPO Data Construction and Training.** To facilitate the DPO training, we construct massive training pairs with positive and negative 3D layouts. Here, positive data stands for the high-quality 3D layouts sourced from our dataset, MesaTask-10K, while the negative data is generated by intentionally corrupting the positive layouts in three distinct ways, each corresponding to a specific shortcoming of our MesaTask model after the SFT. For a 3D layout **L** from our dataset $\mathcal{L}^+$, we randomly select a subset of objects and perturb their positions, rotations, and sizes to deliberately create object collisions, resulting in the negative layout $\mathbf{L}^-_{\mathrm{col}}$ reflecting the collisions. Then, some normal inter-object relations are damaged by altering their relation types, leading to a negative sample $\mathbf{L}^-_{\mathrm{rel}}$ contradicting the task instruction. Finally, we manually remove one or more critical objects to create a negative layout $\mathbf{L}^-_{\mathrm{obj}}$ that neglects task-relevant objects. Therefore, we can obtain a negative dataset $\mathcal{L}^-$ with three distinct layout corruptions. Along with the corresponding task instructions $\mathcal{T}$, we represent the whole paired dataset $\mathcal{D}$ for the DPO training with:

$$\mathcal{D} = (\mathcal{L}^+, \mathcal{L}^-, \mathcal{T}), \qquad \mathcal{L}^- = \{(\mathbf{L}^-_{\mathrm{col}}, \mathbf{L}^-_{\mathrm{rel}}, \mathbf{L}^-_{\mathrm{obj}})\} \text{ for } \mathbf{L} \in \mathcal{L} \tag{4}$$

With the constructed dataset $\mathcal{D}$, we optimize our MesaTask model via the DPO objective following

$$\max_{\pi_\theta} \, \mathbb{E}_{(\mathbf{L}^+, \mathbf{L}^-, \mathbf{T}) \in \mathcal{D}} \log \sigma \left( \beta \log \frac{\pi_\theta \left( \mathbf{L}^+ \mid \mathbf{T} \right)}{\pi_{\mathrm{ref}} \left( \mathbf{L}^+ \mid \mathbf{T} \right)} - \beta \log \frac{\pi_\theta \left( \mathbf{L}^- \mid \mathbf{T} \right)}{\pi_{\mathrm{ref}} \left( \mathbf{L}^- \mid \mathbf{T} \right)} \right), \tag{5}$$

where $\pi_\theta$ is the policy of the fine-tuned LLM, $\sigma(\cdot)$ represents the sigmoid function for preference scoring, and $\beta$ is a temperature parameter that controls the sharpness of the preference margin between positive and negative layouts. With the DPO training, our MesaTask model seeks to acquire a policy $\pi_0$ that favors normal 3D layouts $\mathcal{L}^+$, thereby alleviating three limitations observed in the model after the SFT and consequently enhancing the overall quality of generated tabletop scenes.

## 5 Experiment

### 5.1 Experiment setup

**Dataset.** We build our training data based on the training split of our MesaTask-10k dataset, which contains $10,000$ tabletop scenes. For each scene, we generate five task instructions following the reasoning data creation process above, resulting in a total of $50,000$ task-scene pairs for the supervised fine-tuning. During the stage of DPO training, we construct the paired dataset using $5,000$ previously unseen scenes, where each normal layout sample corresponds to two disrupted layouts on average, thereby yielding a total of $10,000$ positive-negative layout pairs for the DPO training.

**Implementation details.** We adopt Qwen3-8b[31] as the base LLM for both supervised fine-tuning (SFT) and direct preference optimization (DPO). We perform full-parameter fine-tuning in both stages. In the SFT stage, the model is trained for one epoch using the learning rate of $1 \times 10^{-5}$. In the DPO stage, we train for one epoch, with the learning rate of $1 \times 10^{-6}$. All experiments are conducted on a cluster of eight A800 GPUs.

Table 1: **Quantitative comparison** with baseline methods. Our method MesaTask achieves the best generation performance on all evaluation metrics, consistently outperforming other baselines. Meanwhile, we can also observe the performance boost brought by our proposed spatial reasoning chain and the employed DPO training.

| Model | Success Rate(%) | FID↓ | GPT Score | | | | | | User Study |
| --- | --- | --- | --- | --- | --- | --- | --- | --- | --- |
| | | | CwT | OSR | PPI | LCR | OV | Avg. | |
| GPT-4o w/o reason. | 91.6 | 84.3 | 5.02 | 8.04 | 8.90 | 5.74 | 7.05 | 6.95 | 3.11 |
| GPT-4o | 91.4 | 74.4 | 5.30 | 8.06 | 8.99 | 5.96 | 7.12 | 7.09 | 4.21 |
| Holodeck-table | 99.3 | 91.3 | 2.62 | 7.20 | 8.58 | 3.82 | 4.89 | 5.42 | 2.29 |
| I-Design-table | 56.5 | 96.0 | 4.99 | 8.00 | 8.86 | 5.88 | 6.62 | 6.87 | 1.73 |
| Our w/o reason. | **100.0** | 40.8 | 7.14 | 8.59 | 9.13 | 7.48 | 8.66 | 8.20 | 5.43 |
| Ours w/o DPO | 98.4 | 41.4 | 7.18 | 8.59 | 9.15 | 7.52 | **8.71** | 8.23 | 5.75 |
| Ours | 99.1 | **40.3** | **7.22** | **8.64** | **9.17** | **7.53** | **8.71** | **8.25** | **6.12** |

**Baselines.** We evaluate our model against two categories of benchmark methods. The first is closed-source large language models, specifically GPT-4o, where we perform our task in a zero-shot manner. The second category comprises modular scene generation methods, like Holodeck [37] and I-Design[2]. These approaches are originally targeted for indoor scene generation, and are now adapted to fit our task without changing their core frameworks, which are noted as Holodeck-table and I-Design-table here.

**Metrics** We first employ Fréchet Inception Distance (FID) to measure the realism of the generated scenes and the success rate to reflect the syntactic correctness of the LLM-generated output format. A 100% success rate indicates that all the model's outputs are interpretable and can be directly parsed for downstream object retrieval, ultimately enabling the construction of tabletop scenes. Moreover, to conduct a more comprehensive evaluation, we propose the GPT-score, a metric designed to assess the multi-dimensional performance of generated scenes, including Consistency with Task (CwT), Object Size Reasonableness (OSR), Placement Plausibility & Intersections (PPI), Layout Coherence & Realism (LCR), and Object Visibility (OV).

## 5.2 Comparison to baselines

For a fair comparison, each method generates 500 tabletop scenes according to the corresponding task instructions, which will serve as the evaluation corpus for the aforementioned metrics.

**Quantative evaluation.** As shown in Table 1, our method MesaTask demonstrates superior overall performance across all evaluation protocols. In comparison to multiple baseline methods, MesaTask achieves significantly better performance in terms of FID, thereby indicating its capability to generate more realistic tabletop scenes. With respect to the GPT-based multi-dimensional metrics, MesaTask consistently outperforms alternative baseline methods, with particularly notable advantages in CwT and LCR. This superior performance reflects enhanced task-scene alignment and more plausible tabletop layouts, which can be attributed to the high-quality MesaTask-10K dataset we constructed. To assess the perceptual quality of these generated scenes, a total of **127 participants** are invited to conduct a comprehensive user study from three distinct assessment dimensions. The scores presented in Table 1 further confirm that our method achieves the most favorable outcomes in terms of human preference. More evaluation details are put in the supplementary materials.

**Qualitative results** To further substantiate the superior performance of MesaTask, Figure 4 presents qualitative comparisons between our method and three representative baselines. MesaTask consistently produces more realistic and diverse scenes, with a greater number of objects arranged with semantically meaningful and spatially coherent layouts. Moreover, its outputs align more closely with task instructions, capturing nuanced spatial relations such as stacking, containment, and precise object relocation. In contrast, baseline methods often generate overly simplistic or symmetrical layouts, miss key objects, or struggle to interpret complex spatial commands. These qualitative results highlight MesaTask's impressive ability to model task-driven tabletop scenes with plausible layouts that support real-world robotic manipulation.

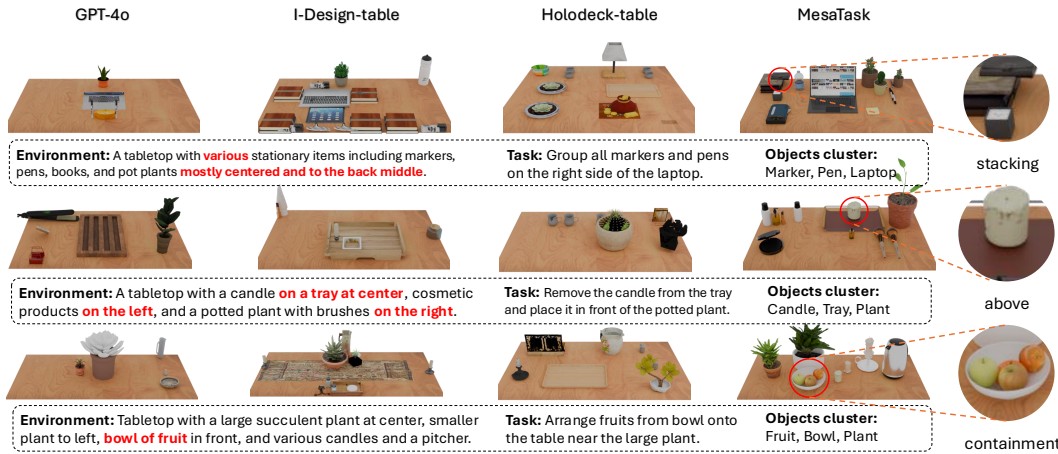

Figure 4: **Qualitative comparison** under the same input task descriptions. Our proposed method, MesaTask, outperforms all baseline approaches across multiple perspectives, specifically exhibiting enhanced realism, superior task-scene alignment, more plausible tabletop layouts, and improved modeling of complex inter-object relationships.

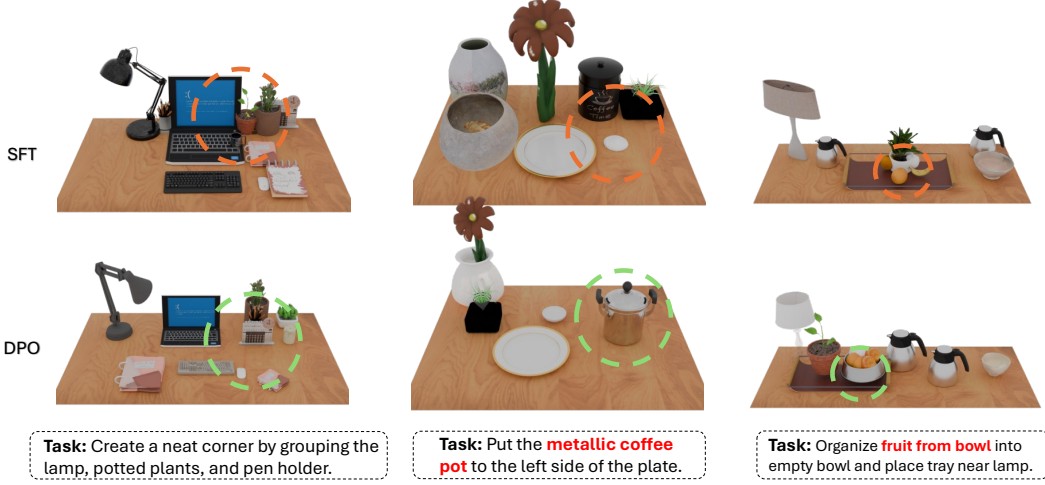

Figure 5: **Ablation study** on DPO training. Compared to the model fine-tuned solely via SFT, the model additionally trained with DPO maintains a lower collision rate (left), higher fidelity to task-related objects (middle), and superior alignment with input task instructions (right).

## 5.3 Ablation study

To better understand the impact of each component in our framework, we conduct a comprehensive ablation study analyzing the contribution of spatial reasoning and preference tuning via Direct Preference Optimization (DPO). Table 1 presents the results of a quantitative ablation study conducted on MesaTask. The removal of either the spatial reasoning module or the DPO training component results in a measurable degradation of MesaTask's overall performance. The qualitative comparisons shown in Figure 5 illustrate the advantages brought by the supplementary DPO training. As observed, DPO effectively alleviates common failure modes exhibited by the SFT-only model, including severe object collisions, the absence of task-relevant objects, and erroneous inter-object relationships. These improvements validate the effectiveness of our adopted DPO training in ensuring coherent table layouts and correct interrelations, leading to realistic tabletop scenes that exhibit greater visual plausibility and enhanced functional fidelity to the given task instructions.

Table 2: **Generalization capability** of MesaTask on four unseen tabletop categories. MesaTask exhibits comparable performance across six distinct evaluation protocols relative to its performance on the test set of the MesaTask-10K dataset.

| Category | Success Rate(%) | GPT Score | | | | | |
|---|---|---|---|---|---|---|---|
| | | CwT | OSR | PPI | LCR | OV | Avg. |
| Nightstand | 100.0 | 7.44 | 8.12 | 9.00 | 7.69 | 8.75 | 8.20 |
| TV stand | 100.0 | 6.56 | 8.06 | 9.06 | 6.94 | 7.44 | 7.61 |
| Side table | 100.0 | 7.62 | 8.62 | 9.25 | 7.69 | 8.50 | 8.34 |
| Cashier counter | 100.0 | 6.25 | 8.38 | 9.06 | 6.62 | 7.50 | 7.56 |

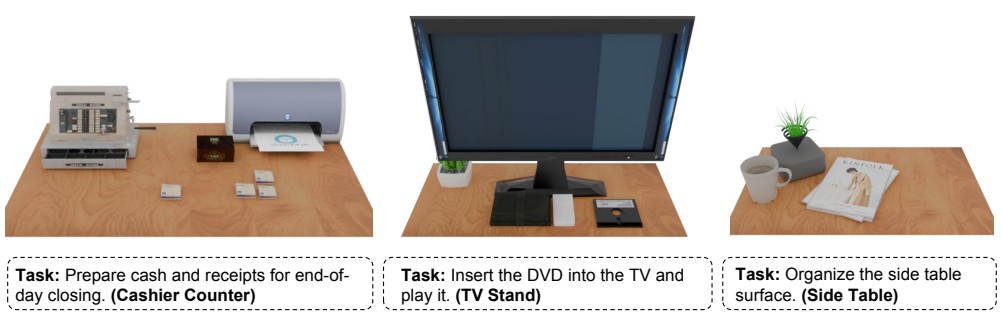

**Task:** Prepare cash and receipts for end-of-day closing. **(Cashier Counter)**  **Task:** Insert the DVD into the TV and play it. **(TV Stand)**  **Task:** Organize the side table surface. **(Side Table)**

Figure 6: MesaTask is capable of generating realistic tabletop scenes belonging to novel categories that are not present in the training dataset.

## 5.4 Generalization capability

To validate the generalization capability of our proposed method, we select tabletop categories not present in MesaTask-10K, including nightstands, TV stands, and side tables from household scenes, as well as cashier counters from shop scenes. For these four tabletop categories, we employ GPT-4o to generate plausible tasks, with 16 tasks assigned to each category. As shown in Table 2, MesaTask exhibits robust generalization capability when tested on the four unseen table categories. The performance of MesaTask across all metrics is comparable to its performance on the test set of six seen categories from MesaTask-10K, as listed in Table 1. Notably, in the case of cashier counters, even though cash registers are not included in MesaTask-10K, our method can accurately generate their descriptions and sizes while placing them correctly. Notably, we can not compute FID since these new scenes are not included in MesaTask-10K. Figure 6 additionally presents several generated tabletop scenes, which belong to these four novel tabletop categories.

## 6 Conclusion

In this paper, we introduce a novel task, namely task-oriented tabletop scene generation, which presents significant challenges owing to the substantial disparity between high-level task instructions and scene layouts. To support this demanding task, we propose a large-scale dataset, **MesaTask-10K**, consisting of roughly $10,700$ tabletop scenes that span six distinct indoor table categories. Thanks to our proposed spatial reasoning chain, our LLM-based framework **MesaTask** will sequentially reason about the scene composition, the spatial interrelationship, the scene graph, and ultimately, the 3D scene layout, based on which 3D assets are retrieved to form a complete tabletop scene. In evaluations, MesaTask demonstrates superior performance over existing baselines in accurately conforming to task instructions and modeling complex inter-object relations. We believe our dataset and framework will inspire a promising research direction and unveil new challenges in this field.

**Limitations and future work**. MesaTask relies on 3D object retrieval, which naturally limits the object diversity to what's available in our 3D object database. In the future, we will explore integrating 3D object generation methods based on bounding box conditions into our tabletop scene generation pipeline. This should allow us to create various objects and more realistic tabletop scenes.

# 7 Acknowledgements

This work is supported in part by the National Key R&D Program of China (2022ZD0160201), Shanghai Artificial Intelligence Laboratory, the National Natural Science Foundation of China (No. 72192821, 62302297, 62472282), Young Elite Scientists Sponsorship Program by CAST (2022QNRC001), and the Fundamental Research Funds for the Central Universities (project number: YG2023QNA35).

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

# Appendix

In this Appendix, we demonstrate the detailed implementation of our main paper. The content of each section is as follows:

- A covers 3D asset annotation, image generation, coarse scene construction, dataset statistics, and benchmark evaluation for MesaTask-10K.

- B explains inference processes, reasoning data construction (including scene graph rules), and DPO data construction for the MesaTask framework.

- C outlines baseline model implementations, evaluation metrics (Success Rate, FID, GPT-Score), and user study details.

- D presents additional qualitative comparisons and further examples of scenes generated by MesaTask.

- E contains all detailed prompts used throughout the methodologies.

## A   Details of MesaTask-10K

### A.1   3D asset annotation via GPT4o

We employ OpenAI's `gpt-4o-mini` to annotate the 3D asset database for MesaTask, aiming to enhance the accuracy of its retrieval and placement results. Inspired by HOLODECK [37] and [42], we render images of an object from orthogonal rotations (0°, 90°, 180°, and 270°) as input, prompting `GPT-4o-mini` to output the following object attributes:

- **Category:** Specifies the precise class to which the object belongs.
- **Description:** Includes details such as the object's name, shape, color, and material.
- **OnTable:** A boolean value indicating whether the object is suitable for placement on a table. For instance, items like chairs or sofas would typically be marked as `false`.
- **Mass:** Represents the mass of the object, preparing this data for potential future physics simulation applications.
- **Front View:** An integer value defining the standard frontal orientation of the object. Objects of the same category should share a consistent front view, which is usually the more symmetrical or informative perspective.
- **IsContainer:** A boolean value determining if the object can hold other items. If this attribute is `true`, the object will possess an "in" relationship when constructing the scene graph, indicating containment.
- **Material:** The material of the 3D objects, such as "wooden", "glass".

### A.2   Tabletop scene image generation stage

For the table type specified by the user, we first use OpenAI's `GPT-4o-mini` to generate five descriptions of the table scene, and then feed prompt text E.1 to GPT.

### A.3   Coarse tabletop scene construction stage

Our process for reconstructing a coarse 3D scene from a single image begins with comprehensive visual understanding. We first employ a Vision Language Model (VLM), specifically `GPT-4o-mini`, to identify the categories of objects in the input image. For each identified object category, Grounded-SAM [22] is utilized to generate precise instance-level segmentation masks. Concurrently, we use DepthAnything V2 [32] to estimate the image's depth map. We subsequently convert this depth map into a 3D point cloud. By combining these instance-level segmentation masks with the overall scene point cloud, we get the point cloud for each object instance. Finally, an axis-aligned bounding box (AABB) is computed for each instance's point cloud, providing an initial layout.

Given the initial layout, we retrieve the 3D asset for each instance. Leveraging the textual descriptions for each object instance obtained from the VLM, we choose the 3D asset that has the highest textual similarity within assets in our database.

Our 3D asset database includes objects curated from Holodeck, the PartNet Mobility dataset, and assets generated by the image-to-3D model (Hunyuan3D), which comprises $11,000$ rigid 3D assets and $1,034$ articulated objects. Objects in PartNet-Mobility are already orientation-aligned. For the other 3D assets, we first centered the camera around the object and uniformly rendered eight views around the z-axis (up). We manually exclude assets that cannot be standardized via z-axis rotation. A VLM model then identifies which of the eight images shows the front view, and the 3D asset is rotated accordingly. After rotating the assets, we manually inspect each object's orientation. Assets not properly rotated to face the front are re-oriented manually. This annotation pipeline ensures all objects in our database are orientation-aligned.

In the coarse placement stage, we aim to resize retrieved 3D assets $o$ to align with the target Axis-Aligned Bounding Box (AABB) $\mathbf{d}' = (w', d', h')$. To this end, we employ the **Isometric Placement** strategy. This strategy places the retrieved 3D assets by uniformly scaling them to preserve their intrinsic XYZ aspect ratio. Specifically, let the dimensions of a retrieved asset $o$ be $\mathbf{d} = (w, d, h)$. The scaling factor, $s$, is calculated by resizing the shortest dimension of the asset (along its X, Y, or Z principal axes) to match the length of the corresponding dimension of the target instance's AABB $\mathbf{d}'$. For example, if $w$ is the shortest dimension of $\mathbf{d}$, then $s = w'/w$. All other asset dimensions are then scaled by the same factor $s$ to maintain their original aspect ratio, resulting in scaled dimensions $s \cdot \mathbf{d} = (s \cdot w, s \cdot d, s \cdot h)$. This particular approach to scaling is predicated on the inherent limitations of single-view depth estimation; challenges such as partial occlusions and errors in predicted depth values frequently compromise the reliability of these image-derived AABB dimensions $\mathbf{d}'$, making a cautious, proportion-preserving scaling method essential. Preserving the asset's relative XYZ proportions in this manner is also crucial for facilitating subsequent manual annotation and refinement. Furthermore, it is important to note that the aim of this coarse placement stage is not to achieve perfect 3D reconstruction but rather to generate approximate bounding box data $s \cdot \mathbf{d}$ sufficiently aligned with the scene to serve as effective training data for later stages.

## A.4 The effort of human annotators

The coarse construction stage is not sufficiently plausible due to challenges such as occlusion, inaccurate depth estimation, and retrieval errors. Thus, human annotators played a critical and extensive role in refining the layout. Specifically, for each sample, annotators were provided with: the GLB file of the tabletop scene, the rendered scene image, as well as individual object snapshots and their indices. Using Blender, annotators manually adjusted the scene layout to match the reference image, which involves several steps: 1) Translating objects to correct positions, 2) Scaling them to appropriate sizes, 3) Rotating them to align orientations correctly, 4) Ensuring spatial plausibility of inter-object relations. The annotation complexity varied substantially depending on the number of objects and the complexity of their spatial configurations. For example, scenes with many small objects and dense relations were significantly more complex and time-consuming. On average, annotators spend 10 to 20 minutes per scene.

After receiving each annotated 3D file from annotators, we render all annotated 3D scenes from four directions: front, back, left, and right. These renderings are compared with reference images. If quality fails to meet requirements, such as frequent issues like unreasonable object sizes or objects floating in the air, we request annotators to revise the errors until they meet the acceptance criteria.

## A.5 Dataset statistics

Our 3D asset database has an extensive collection of over 200 common tabletop object categories, which has numerous high-fidelity 3D models as shown in Figure 7.

## A.6 Tabletop scene generation benchmark

In Table 3, we introduce Mesatask-10k as a new benchmark dataset to evaluate scene generation methods across five representative tabletop environments: Coffee Table, Dining Table, Dressing Table, Kitchen Counter, and Office Table. There is no comparison with the bathroom vanity because

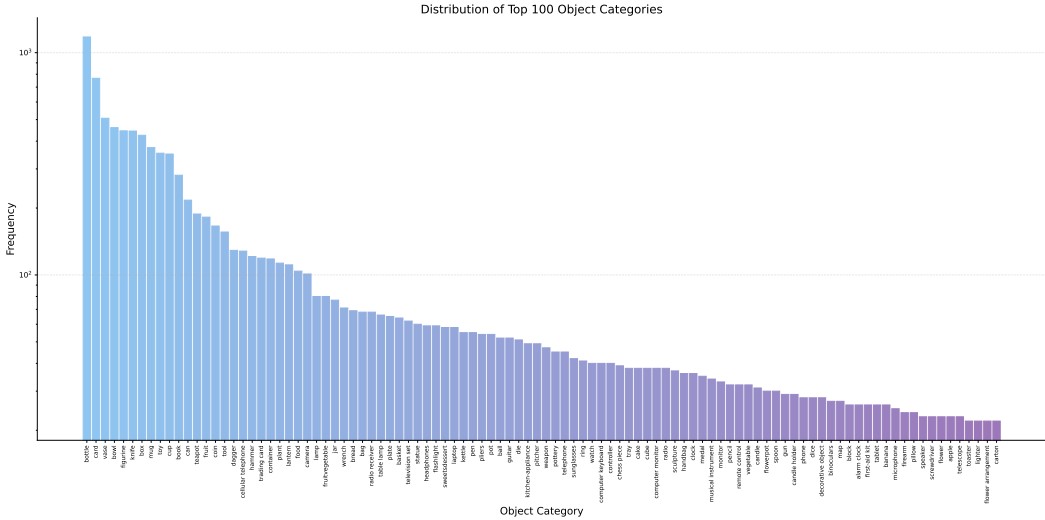

Figure 7: Distribution of the top 100 object categories

Table 3: Comparison of ATISS, DiffuScene, and PhyScene trained on **MesaTask-10k**.

| Scene | ATISS [20] | | | DiffuScene [25] | | | PhyScene [34] | | |
|---|---|---|---|---|---|---|---|---|---|
| | FID ↓ | KID ↓ | CKL ↓ | FID ↓ | KID ↓ | CKL ↓ | FID ↓ | KID ↓ | CKL ↓ |
| **Coffee Table** | 41.85 | 0.0101 | 0.217 | 40.36 | 0.0091 | 0.224 | 43.41 | 0.0137 | 0.247 |
| **Dining Table** | 59.89 | 0.0291 | 0.806 | 53.61 | 0.0185 | 1.051 | 50.13 | 0.0131 | 0.790 |
| **Dressing Table** | 42.07 | 0.0103 | 0.685 | 45.95 | 0.0117 | 0.772 | 39.54 | 0.0098 | 0.464 |
| **Kitchen Counter** | 51.81 | 0.0232 | 1.229 | 51.49 | 0.0194 | 1.231 | 50.07 | 0.0182 | 0.756 |
| **Office Table** | 59.22 | 0.0313 | 0.217 | 42.47 | 0.0118 | 0.224 | 35.09 | 0.0047 | 0.357 |

there is a pool in the bathroom vanity, and these methods cannot deal with this problem. The methods assessed include ATISS [20], DiffuScene [25], and PhyScene [34]. Evaluation metrics comprise FID (Fréchet Inception Distance), KID (Kernel Inception Distance), and CKL (Category KL Divergence), with lower values indicating better performance. The visualization results of these methods are shown in Figure 8

ATISS, while slightly lagging in FID and KID, shows consistent results across all scenes, indicating stability in its generative quality. PhyScene exhibits relatively higher FID and KID in simpler layouts but achieves competitive scores in more complex scenes such as Kitchen Counter and Dressing Table, likely due to its integration of physical constraints that enhance realism in interaction-heavy settings.

The CKL metric reveals complementary insights. Lower CKL values indicate that a model captures the categorical distribution of objects more faithfully. ATISS and PhyScene generally achieve lower CKL scores in scenes with moderate complexity, suggesting a more accurate modeling of object co-occurrence and diversity. DiffuScene, despite strong performance in FID and KID, often yields higher CKL values, particularly in scenes such as Kitchen Counter and Dining Table, where a large number of distinct object categories are present. This discrepancy suggests that DiffuScene may prioritize visual plausibility over accurate semantic diversity, whereas ATISS and PhyScene better balance both.

Overall, the results highlight a trade-off between visual quality and semantic alignment. While DiffuScene excels in producing visually coherent layouts, ATISS and PhyScene demonstrate stronger alignment with real-world category distributions. The inclusion of CKL as an evaluation metric in Mesatask-10k proves critical for revealing these nuances, emphasizing the importance of considering object diversity and distribution fidelity alongside traditional image-level metrics.

It's noteworthy that our method, MesaTask, is quite different from these three representative scene generation methods. Our approach focuses on generating scenes from task instructions, whereas prior methods such as ATISS, DiffuScene, and PhyScene generate scenes from simple scene descriptions.

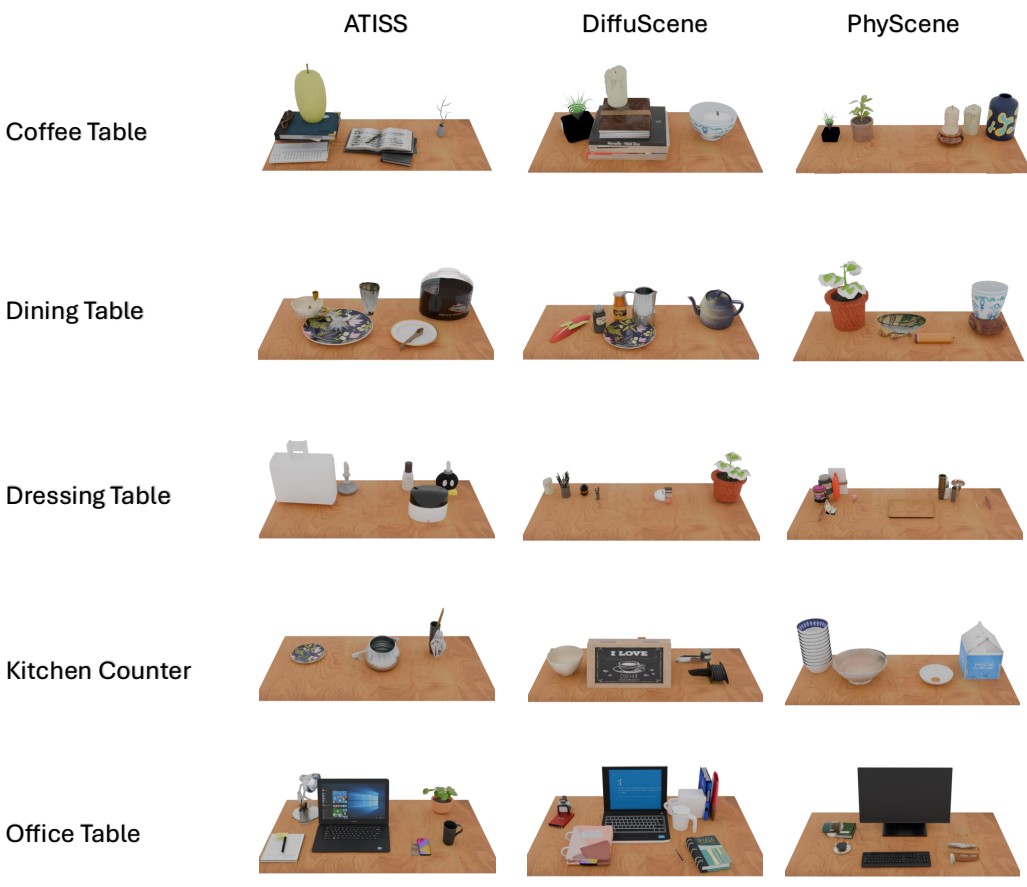

Figure 8: Qualitative results of ATISS, DiffuScene, and PhyScene Trained on MesaTask-10k

This fundamental difference in settings precludes direct comparisons. Moreover, MesaTask is capable of generating open-vocabulary tabletop scenes given task instructions. Leveraging a fine-tuned LLM to generate scene layouts, it can produce layouts and textual descriptions for objects not present in the training set. In contrast, ATISS, DiffuScene, and PhyScene generate predefined object features and cannot generate objects out of the training set at inference time.

# B    Details of MesaTask

## B.1    Inference details

During the inference phase, the user input consists of three parts: 1. Task Instruction, which is typically a description of the task (e.g., "Organize fruit from bowl into empty bowl and place tray near lamp"); 2. Size of the tabletop surface. The tabletop surface is specified as a list $[x_{min}, y_{min}, x_{max}, y_{max}]$ in centimeters (cm), defining a rectangular area, which is the object's place. The user input is then processed via the OpenAI GPT-4o API to generate the detailed task information with prompt E.2. 3. Size of the No Placement Area (Optional): a list $[x_{min}, y_{min}, x_{max}, y_{max}]$ in centimeters (cm), defining a rectangular area that objects can not be placed, in the table type like bathroom vanity, the this area refer to the sink area.

Our **MesaTask** model subsequently generates tabletop scene layouts from the detailed task information. Its primary outputs encompass two key components: a *Spatial Reasoning Chain*—which includes stages such as Object Completion, Interrelation Inference, and Scene Graph generation—and ultimately, the final tabletop scene layout. This layout specifies the position and size of the objects.

Given the generated scene layout, we retrieve the objects from the 3D assets database and place them on the tabletop. Our 3D asset database comprises objects $o$, each defined by a tuple $o = (t, \mathbf{d})$, where $t$ is a textual description (e.g., "red coffee mug") and $\mathbf{d} = (w, d, h)$ represents the dimensions (width, depth, height) of its 3D bounding box. Correspondingly, each object $o'$ in the generated scene layout is characterized by its target 3D bounding box size $\mathbf{d}' = (w', d', h')$, its desired position $\mathbf{p}' = (x', y', z')$, and an associated textual description $t'$.

Inspired by HOLODECK [37], the selection of the most appropriate 3D asset from the library for each target object $o'$ is guided by two metrics: Text Similarity and Size Similarity. Text similarity, $T(t, t')$, between the asset description $t$ and the target description $t'$ is computed using a Sentence Transformer (SBERT), specifically the `all-mpnet-base-v2` checkpoint: $T(t, t') = \text{SBERT}(t, t')$.

Complementing textual matching, Size Similarity addresses the practical consideration that a single textual description can correspond to objects of various sizes. For this, we represent the size of the asset $o$ as a vector $\mathbf{d}$ and that of the target object $o'$ as $\mathbf{d}'$. The size similarity, $S(o, o')$, is then computed using the cosine similarity as: $S(o, o') = \cos(\mathbf{d}, \mathbf{d}')$.

The final retrieval score $R(o, o')$ for a library asset $o$ with respect to a target object $o'$ is formulated as a weighted sum of these two similarities:

$$R(o, o') = \alpha \cdot T(t, t') + \beta \cdot S(o, o')$$

The weights are empirically set to $\alpha = 0.9$ and $\beta = 0.1$, prioritizing textual relevance while still accounting for dimensional congruence. The 3D asset that yields the highest retrieval score $R(o, o')$ is then selected for subsequent placement into the scene according to $\mathbf{p}'$ and size $\mathbf{d}'$.

## B.2 Reasoning data construction

For a scene $\mathbf{S}$ in MesaTask-10K, we first extract the scene graph by following the rule-based method below. To compensate for the spatial relations missing in the scene graph, we utilize GPT-4o [1] with *Table Description* prompt E.6 to output a detailed scene description $\mathbf{D}$ based on the rendered tabletop scene image. Given the scene graph $\mathcal{G}_{\mathbf{S}}$ and scene descriptions $\mathbf{D}$, the multimodal LLM is prompted to generate a complete object list $\mathcal{V}$ and inter-object relations $\mathcal{E}$, as well as the corresponding task instructions $\mathbf{T}$, in particular including aforementioned detailed task information with *Task from Scene Graph* prompt and *Reasoning Context Generation* prompt E.6.

**Scene graph extraction.** We define a set of interpretable geometric rules grounded in relative positions, distances, and orientations, as shown in Table4. The Left of, Right of, In front of, and Behind relationships are determined based on the relative x and y position differences between object centroids, constrained by a distance threshold proportional to the table size. Vertical relationships, including above and below, are defined using the z-coordinate range of objects and require sufficient horizontal overlap to ensure meaningful interaction. The In relationship captures containment, requiring both high horizontal and vertical overlap ratios.

Object orientation is characterized by the Face to relations, which map the z-axis rotation angle of an object to eight discrete directional bins, such as Front, Back, Left, and the diagonals. Positional context is described by the Is at relation, which locates an object within a 3×3 spatial grid overlaid on the table surface. Finally, the Are equally spaced along rule detects linear arrangements of three or more objects with approximately uniform spacing along either the x- or y-axis, within a 10% tolerance range.

## B.3 DPO data construction

To facilitate training via Direct Preference Optimization (DPO), we construct a dataset of preference-labeled 3D scene layout samples. Each sample consists of a natural language prompt along with a pair of completions: a *positive* layout that is preferred, and a *negative* layout that is dispreferred. These pairs are used to model the implicit preferences between layout candidates under the same instruction, allowing the DPO algorithm to learn alignment signals from relative quality judgments.

The positive samples are drawn from our curated dataset, **MesaTask-10K**, which contains high-quality 3D layouts that successfully fulfill the spatial requirements of the associated instructions. Each entry includes a reasoning process (referred to as "thinking process") and an output layout represented in structured JSON format, including geometric attributes such as object positions, rotations, and

Table 4: Rules to determine the spatial relationships between objects.

| Relationship | Rule |
|---|---|
| Left of | $|dx| > |dy|$ and $dx < 0$ and $d(s,o) \leq 0.4 \cdot \max(table\_size)$ |
| Right of | $|dx| > |dy|$ and $dx > 0$ and $d(s,o) \leq 0.4 \cdot \max(table\_size)$ |
| In front of | $|dy| > |dx|$ and $dy < 0$ and $d(s,o) \leq 0.4 \cdot \max(table\_size)$ |
| Behind | $|dy| > |dx|$ and $dy > 0$ and $d(s,o) \leq 0.4 \cdot \max(table\_size)$ |
| Above | $z\_min_1 > z\_max_2 - threshold$ and $overlap_{horizontal} \geq 0.5$ |
| Below | $z\_max_1 < z\_min_2 + threshold$ and $overlap_{horizontal} \geq 0.5$ |
| In | $overlap_{horizontal} \geq 0.9$ and $overlap_{vertical} \geq 0.5$ |
| Face to front | $-\frac{\pi}{8} \leq \theta < \frac{\pi}{8}$ |
| Face to front_right | $\frac{\pi}{8} \leq \theta < \frac{3\pi}{8}$ |
| Face to right | $\frac{3\pi}{8} \leq \theta < \frac{5\pi}{8}$ |
| Face to back_right | $\frac{5\pi}{8} \leq \theta < \frac{7\pi}{8}$ |
| Face to back | $\frac{7\pi}{8} \leq \theta$ or $\theta < -\frac{7\pi}{8}$ |
| Face to back_left | $-\frac{7\pi}{8} \leq \theta < -\frac{5\pi}{8}$ |
| Face to left | $-\frac{5\pi}{8} \leq \theta < -\frac{3\pi}{8}$ |
| Face to front_left | $-\frac{3\pi}{8} \leq \theta < -\frac{\pi}{8}$ |
| Is at | Relative position in 9-grid division of table |
| Are equally spaced along | Three or more objects with equal spacing in X or Y direction |

sizes, as well as an explicit symbolic scene graph describing inter-object spatial relationships (e.g., "(Cup, left of, Bowl)").

To obtain corresponding negative samples, we apply one of the following three corruption strategies to the original layout, each designed to reflect a typical failure mode observed in model outputs after supervised fine-tuning (SFT):

- **Geometric Perturbation (Collision Induction):** A subset of objects in the layout is selected at random, and their spatial attributes (position, rotation, or size) are perturbed. The perturbations are bounded to remain within the layout's item placement zone but are large enough (e.g., up to 20% of the region's width or height) to induce spatial conflicts or object collisions. The type of perturbation is chosen probabilistically, favoring position changes (e.g., with 80% probability). The resulting layout $\mathcal{L}_{col}^{-}$ often violates basic physical plausibility.

- **Scene Graph Corruption (Semantic Misalignment):** The reasoning trace, particularly the scene graph, is altered by either removing a subset of the spatial relations or replacing them with incorrect ones. For example, "(Book, on, Shelf)" might be replaced with "(Book, under, Shelf)." This yields a logically inconsistent reasoning trace, denoted as $\mathcal{L}_{rel}^{-}$, which conflicts with the spatial semantics of the original instruction.

- **Object Removal (Functional Deficiency):** One or more task-relevant objects are randomly deleted from the layout. This operation simulates incomplete or underspecified layouts, producing samples denoted by $\mathcal{L}_{miss}^{-}$ that are geometrically valid but semantically deficient with respect to the task requirements.

For each instruction, a prompt is constructed by concatenating the instruction and input fields. The **chosen** completion consists of the original reasoning trace followed by the correct layout output. The **rejected** completion is created by applying one of the aforementioned corruption strategies. To increase supervision signal diversity, we sample two independent rejected completions per prompt using different corruption methods or random seeds.

# C Details of experiment

## C.1 Implementation of baseline model

**Holodeck-Table**   HOLODECK[37] is an excellent method for generating scene layouts. It leverages a commercial, closed-source large language model (GPT) to first generate the number, types, and spatial relationships of objects in a scene. Then, it retrieves appropriate 3D assets and uses an optimization-based search algorithm to place them plausibly within the environment.

We believe this layout generation approach can be adapted to desktop environments, so we implemented a modified version tailored for desktop object placement, which we call Holodeck-Table. A key aspect of leveraging large language models lies in prompt engineering. Building upon the original HOLODECK prompts, we designed prompts more suited to desktop scenarios and removed modules irrelevant to desktop layouts, such as the Wall Module and Window Module. The modified prompt is shown in E.4.

However, prompt modification alone is insufficient to generate reasonable desktop layouts. This is due to fundamental differences between room-scale and desktop-scale environments. For example, furniture tends to be placed near walls, while desktop objects are often positioned away from the table edges. Therefore, we also modified HOLODECK's optimization phase to better accommodate desktop scene constraints and ensure the generated layouts are as realistic as possible.

**I-Design-Table**   I-Design[2] enables users to easily express their interior design preferences through natural language interaction, and transforms those preferences into visualized 3D layouts. The system employs a set of large language model agents that communicate and reason with each other to convert user input into a feasible scene graph, establishing the relative spatial relationships between objects. After generating the scene graph, I-Design uses a backtracking algorithm to determine the optimal placement of each object in the scene. Owing to the relatively simple design of I-Design's optimization stage, no substantial modifications to the underlying algorithm were required. By appropriately adapting the original prompt, we developed I-Design-Table, which is capable of generating semantically and spatially coherent desktop object arrangements.The modified prompt is shown in the E.5.

## C.2 Details of metrics

**Success Rate**   For LLM-based methods, we define a response as successful if it adheres to the expected output format and includes at least one valid object. The success rate is computed as the ratio of successful responses to the total number of test cases.

**Fréchet Inception Distance (FID)**   We adopt Fréchet Inception Distance (FID) to evaluate the visual realism of rendered tabletop scenes. FID compares the distribution of deep features extracted from generated images against those from ground-truth images, reflecting perceptual similarity at a high semantic level. Specifically, we render both real and generated 3D scenes from a front-facing camera view to obtain consistent 2D images. Each image is resized to $299 \times 299$, normalized to the $[-1, 1]$ range, and converted to RGB (with alpha-composited black background if needed). We extract 2048-dimensional feature vectors using a pretrained Inception-V3 [23] network, where the classification head is replaced with an identity mapping to preserve penultimate-layer activations.

Let $\mathcal{X}_r$ and $\mathcal{X}_g$ be the feature sets extracted from real and generated images, with empirical means $(\mu_r, \mu_g)$ and covariances $(\Sigma_r, \Sigma_g)$, respectively. FID is computed as:

$$\text{FID} = \|\mu_r - \mu_g\|_2^2 + \text{Tr}(\Sigma_r + \Sigma_g - 2(\Sigma_r \Sigma_g)^{1/2}). \tag{6}$$

To ensure numerical stability, we follow standard practices: the matrix square root is computed via the SciPy `sqrtm` function, and only the real part is retained. When necessary, a small regularization term is added to the diagonal of covariance matrices to ensure positive semi-definiteness. FID in our setting serves as a quantitative proxy for how photorealistic and coherent the generated 3D scenes appear when rendered to 2D.

**GPT-Score**   To evaluate the semantic alignment and perceptual quality of generated 3D tabletop scenes, we propose a multi-dimensional assessment protocol based on GPT-based scoring [36, 24]. A

## Scene #148

**Environment:**

A tabletop with a closed laptop, spiral notebook, scissors, mouse, water dispenser, pen holder, and thermos.

**Task:**

Prepare the workspace for note-taking by clearing the mouse and opening the laptop.

**Goal:**

- Pick up the mouse and place it near the thermos on the far left
- Open the laptop lid
- Pick up the notebook and place it directly in front of the laptop

**Objects cluster:**

- Mouse
- Thermos
- Laptop
- Notebook

**Rating Criteria:**

- **Realism & Visual Quality**: The overall visual quality and realism of the image (1=Poor, 7=Excellent)
- **Alignment with Text & Task**: How well the image matches the scene description (1=Poor, 7=Excellent)
- **Spatial Coherence**: Whether the spatial layout and relationships between objects in the scene are reasonable (1=Poor, 7=Excellent)

**Please ensure your ratings have variance and don't give all images the same score, aim for the greatest possible differentiation between images.**

## Image #1

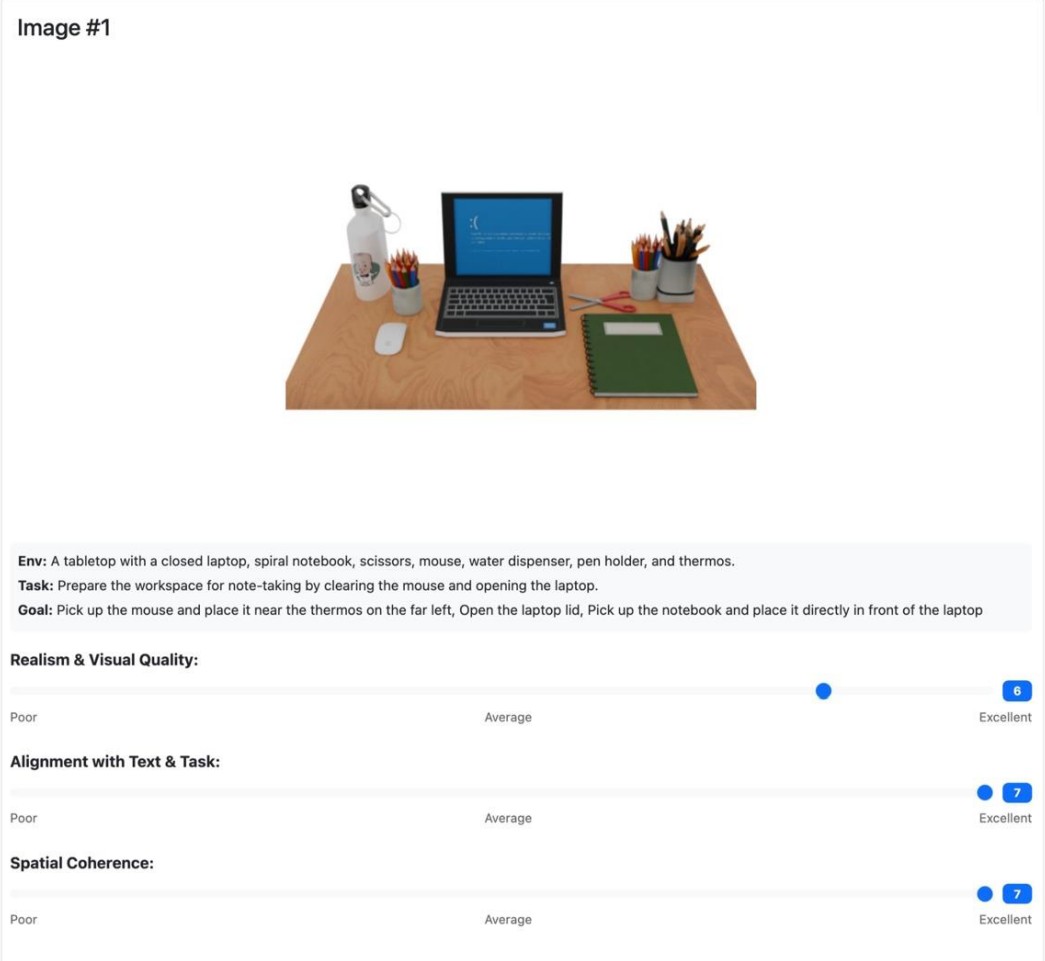

**Env:** A tabletop with a closed laptop, spiral notebook, scissors, mouse, water dispenser, pen holder, and thermos.
**Task:** Prepare the workspace for note-taking by clearing the mouse and opening the laptop.
**Goal:** Pick up the mouse and place it near the thermos on the far left, Open the laptop lid, Pick up the notebook and place it directly in front of the laptop

**Realism & Visual Quality:**

| Poor | Average | 6 | Excellent |

**Alignment with Text & Task:**

| Poor | Average | 7 | Excellent |

**Spatial Coherence:**

| Poor | Average | 7 | Excellent |

Figure 9: User study interface

pretrained large language model is prompted to analyze both the rendered front-view and perspective-view images of each scene, along with the corresponding task description, including environment and task, and to rate the layout across five distinct criteria:

- **Consistency with Task**: Whether the scene's object composition and spatial arrangement are coherent with the task description.
- **Object Size Reasonableness**: Whether object sizes are proportionate and physically realistic.
- **Placement Plausibility & Intersections**: Whether objects are grounded naturally without unrealistic interpenetrations.
- **Layout Coherence & Realism**: Whether the overall layout is visually functional, realistic, and context-appropriate.
- **Object Visibility**: Whether key task-relevant objects are clearly visible and identifiable in at least one view.

Each criterion is scored on a scale from 1 (poor) to 10 (excellent), and includes a short explanation to justify the score. The evaluation is carried out via a structured prompt specifically designed to elicit detailed, consistent assessments across scenes. For full prompt details, please refer to Section E.3.

### C.3 User study

To further assess the perceptual quality and human preference of generated tabletop scenes, we conducted a user study based on rendered scene images. Participants were asked to rate scenes across three key dimensions:

- **Realism & Visual Quality**: The overall visual realism and aesthetic fidelity of the image. (1 = Poor, 7 = Excellent)
- **Alignment with Text & Task**: The degree to which the scene matches the given task description and aligns with the intended semantic content. (1 = Poor, 7 = Excellent)
- **Spatial Coherence**: Whether the spatial arrangement and inter-object relationships in the scene are logical and physically plausible. (1 = Poor, 7 = Excellent)

Each rendered scene is rated on a 7-point Likert scale for the above criteria, and the final score for a scene is computed as the average of the three dimension scores.

Our custom user study interface (see Figure 9) randomly samples 5 scenes per participant from the test set. For each scene, the interface displays images generated by different methods in a randomly shuffled order to mitigate position bias. Each image is evaluated independently without revealing the identity of the generating method. A total of **127 participants** completed the study, resulting in a diverse and robust set of human evaluations for comparative analysis across methods.

## D More result

Inspired by ManiTaskGen [5], we further investigate our model's performance across tasks with varying complexity levels. Accordingly, we categorized tasks of varying types and complexity levels into four task difficulty levels, specifically:

- Level 1: Single-step pick-and-place tasks with a unique target object and no perceptual ambiguity (e.g., "Move the red dictionary on the bookshelf to the table");
- Level 2: Single-step pick-and-place tasks with non-unique target objects, requiring additional descriptions for distinction (e.g., "Move the blue cup on the table to the coffee table" where multiple cups exist in the scene);
- Level 3: Multi-step tasks formed by two Level 1 or Level 2 tasks connected by "THEN" (e.g., "First move the book from the bookshelf to the left of the table, then move it to the right of the table");
- Level 4: Outcome-based abstract tasks describing the target scene state rather than specific steps (e.g., "Tidy up the messy desk", "Make the living room cleaner").

Table 5: **Quantitative performance results** of MesaTask are presented across tasks with varying types and complexity levels.

| Level | Success Rate(%) | FID↓ | GPT Score | | | | | |
|---|---|---|---|---|---|---|---|---|
| | | | CwT | OSR | PPI | LCR | OV | Avg. |
| level1 | 99.8 | 59.3 | 7.20 | 8.88 | 9.40 | 7.22 | 7.80 | 8.10 |
| level2 | 99.4 | 55.5 | 7.44 | 8.40 | 9.36 | 7.44 | 8.16 | 8.16 |
| level3 | 99.2 | 50.9 | 7.04 | 8.36 | 9.46 | 7.28 | 8.10 | 8.05 |
| level4 | 98.4 | 43.9 | 7.46 | 8.78 | 9.70 | 7.68 | 8.88 | 8.50 |

Table 6: MesaTask generates collision-free scenes after physics-based post-processing, satisfying physical plausibility.

| Method | GPT-4o | I-Design-table | Holodeck-table | MesaTask w/o simulation | MesaTask |
|---|---|---|---|---|---|
| Col. Rate(%) | 21.13 | 9.92 | **0** | 11.21 | **0** |

We provided the definitions of these four task levels to GPT-4o to generate diverse tasks, yielding 500 tasks per level. Each level's tasks cover six common indoor tabletops (bathroom vanity, dining table, kitchen counter, coffee table, dressing table, office table). We evaluated the generated scenes using the same metrics as in the main paper. As shown in Table 5, tabletop scenes generated under all task levels achieved high scores in the multi-dimensional assessment, confirming that our method can effectively handle tasks of varying types and complexity levels.

However, we observed variations in FID across different task levels. This discrepancy arises because task instructions in the training set are predominantly at Level 4 difficulty (when training set tasks were classified using GPT-4o according to the above criteria, $83.8\%$ fell into Level 4, $11.4\%$ into Level 3, $3.8\%$ into Level 2, and $1\%$ into Level 1).

To calculate physical plausibility metrics, we thus evaluate occupancy overlaps between objects rather than bounding box intersections. Specifically, the layout of each scene, along with corresponding object assets, is transformed into Drake's[26] internal scene tensor representation. We then use Drake's geometry engine to compute signed distances between all pairs of collision geometries. A negative signed distance indicates interpenetration, which is counted as a collision event. We obtained physical plausibility results on the test set in the main paper, with the collision rate metric defined as the number of collision object pairs $N_{collision}$ divided by the total number of potentially collision object pairs $N_{total}$. The quantitative comparison is listed in Table 6.

We provide more qualitative comparisons between our method MesaTask and existing baselines, including GPT-4o [1], I-Design-table [37], and Holodeck-table. As shown in Figure 10, each row shows a task-conditioned tabletop scene generated by different methods under the same instructions, illustrating their differences in object selection and arrangement. We also include additional qualitative results generated by our MesaTask model in Figure 11, Figure 12, and Figure 13. These examples show how MesaTask handles various task types such as cleaning, organizing, and preparing tabletop scenes. In the figures, the pink flat squares represent the sink area of a bathroom vanity, which is treated as a no-placement zone.

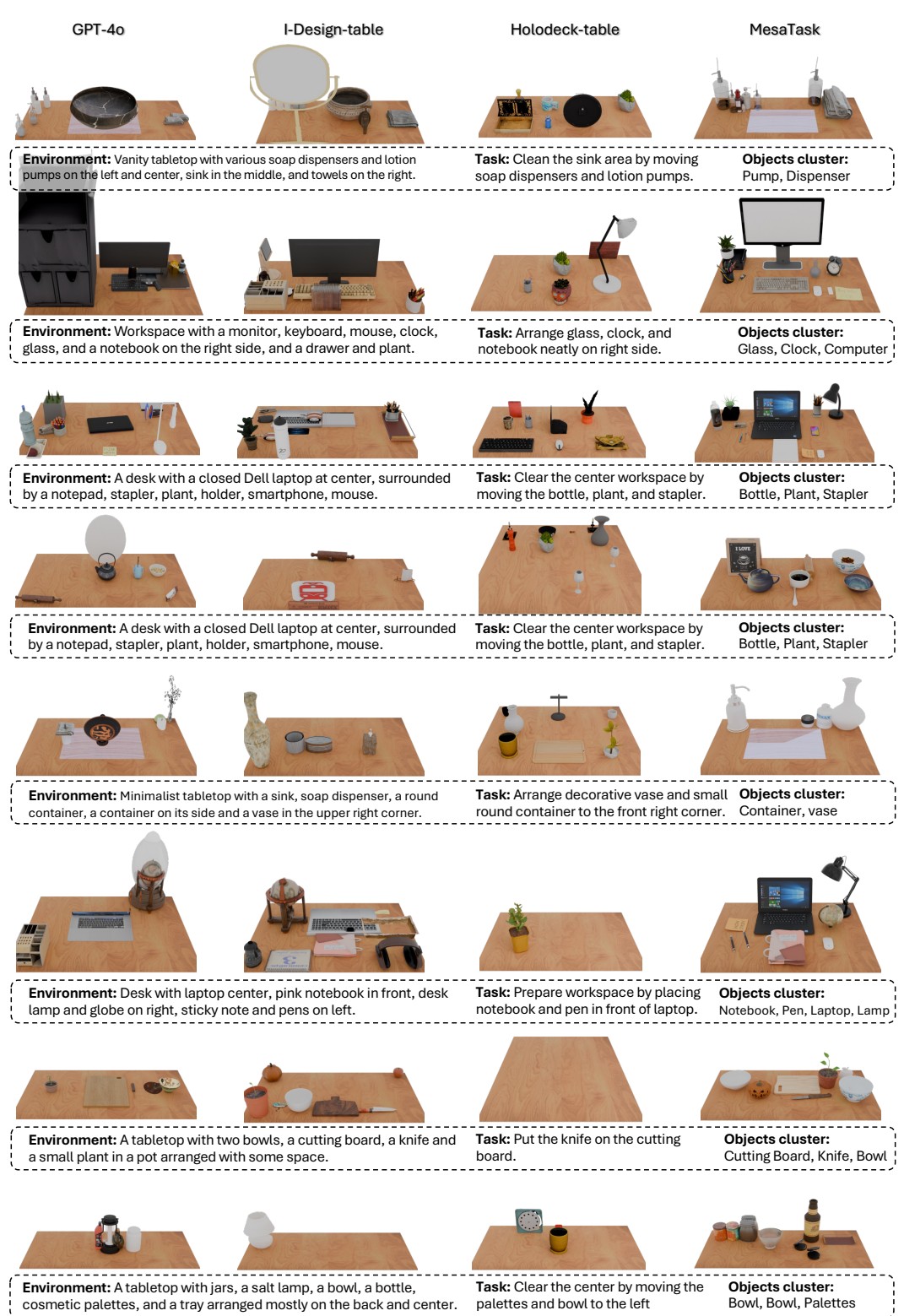

GPT-4o    I-Design-table    Holodeck-table    MesaTask

**Environment:** Vanity tabletop with various soap dispensers and lotion pumps on the left and center, sink in the middle, and towels on the right.

**Task:** Clean the sink area by moving soap dispensers and lotion pumps.

**Objects cluster:** Pump, Dispenser

**Environment:** Workspace with a monitor, keyboard, mouse, clock, glass, and a notebook on the right side, and a drawer and plant.

**Task:** Arrange glass, clock, and notebook neatly on right side.

**Objects cluster:** Glass, Clock, Computer

**Environment:** A desk with a closed Dell laptop at center, surrounded by a notepad, stapler, plant, holder, smartphone, mouse.

**Task:** Clear the center workspace by moving the bottle, plant, and stapler.

**Objects cluster:** Bottle, Plant, Stapler

**Environment:** A desk with a closed Dell laptop at center, surrounded by a notepad, stapler, plant, holder, smartphone, mouse.

**Task:** Clear the center workspace by moving the bottle, plant, and stapler.

**Objects cluster:** Bottle, Plant, Stapler

**Environment:** Minimalist tabletop with a sink, soap dispenser, a round container, a container on its side and a vase in the upper right corner.

**Task:** Arrange decorative vase and small round container to the front right corner.

**Objects cluster:** Container, vase

**Environment:** Desk with laptop center, pink notebook in front, desk lamp and globe on right, sticky note and pens on left.

**Task:** Prepare workspace by placing notebook and pen in front of laptop.

**Objects cluster:** Notebook, Pen, Laptop, Lamp

**Environment:** A tabletop with two bowls, a cutting board, a knife and a small plant in a pot arranged with some space.

**Task:** Put the knife on the cutting board.

**Objects cluster:** Cutting Board, Knife, Bowl

**Environment:** A tabletop with jars, a salt lamp, a bowl, a bottle, cosmetic palettes, and a tray arranged mostly on the back and center.

**Task:** Clear the center by moving the palettes and bowl to the left

**Objects cluster:** Bowl, Bowl, Palettes

Figure 10: More qualitative comparisons of task-conditioned scene generation results across GPT-4o, I-Design-table, Holodeck-table, and our proposed MesaTask.

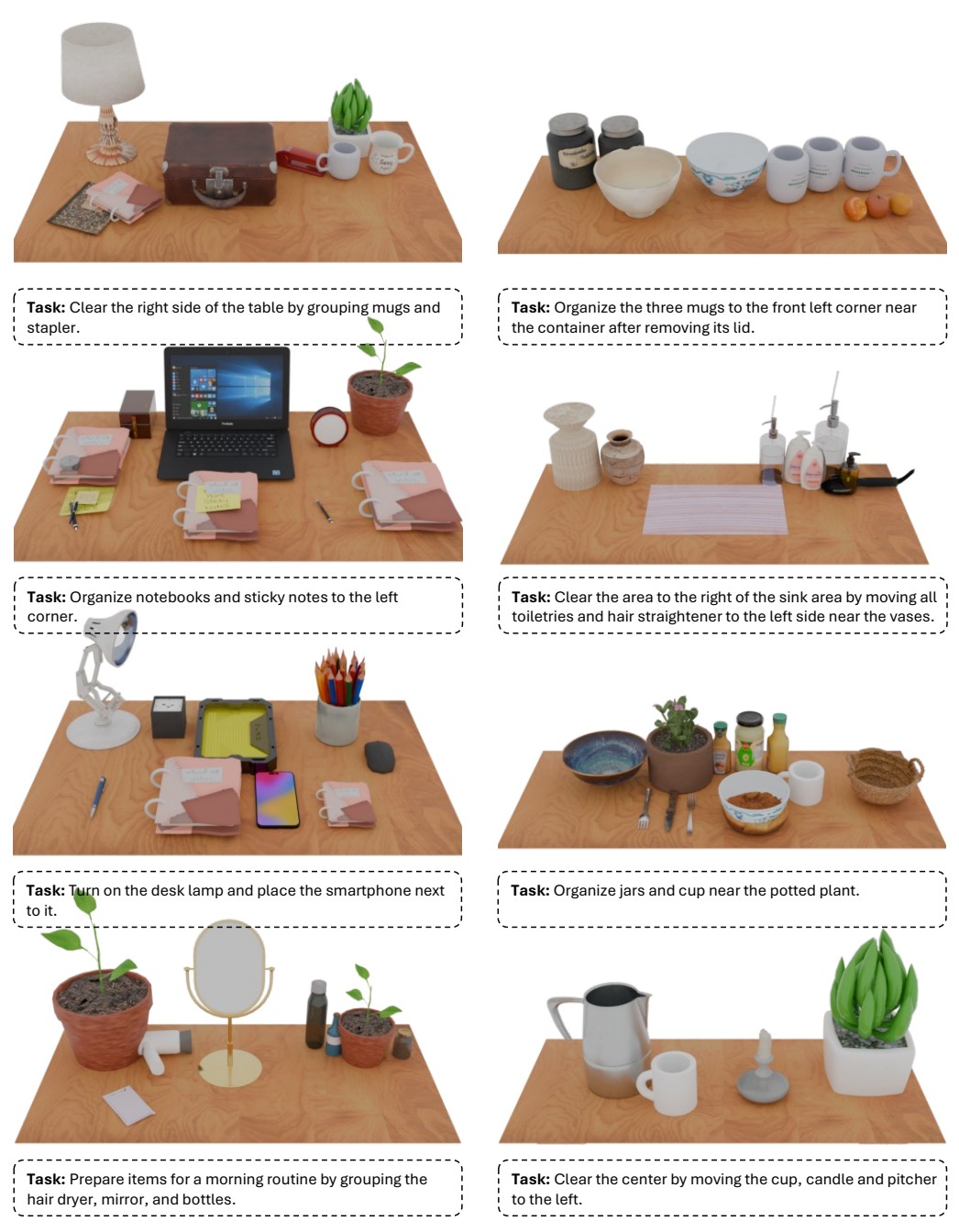

**Task:** Clear the right side of the table by grouping mugs and stapler.

**Task:** Organize the three mugs to the front left corner near the container after removing its lid.

**Task:** Organize notebooks and sticky notes to the left corner.

**Task:** Clear the area to the right of the sink area by moving all toiletries and hair straightener to the left side near the vases.

**Task:** Turn on the desk lamp and place the smartphone next to it.

**Task:** Organize jars and cup near the potted plant.

**Task:** Prepare items for a morning routine by grouping the hair dryer, mirror, and bottles.

**Task:** Clear the center by moving the cup, candle and pitcher to the left.

Figure 11: Additional qualitative results generated by our proposed MesaTask.

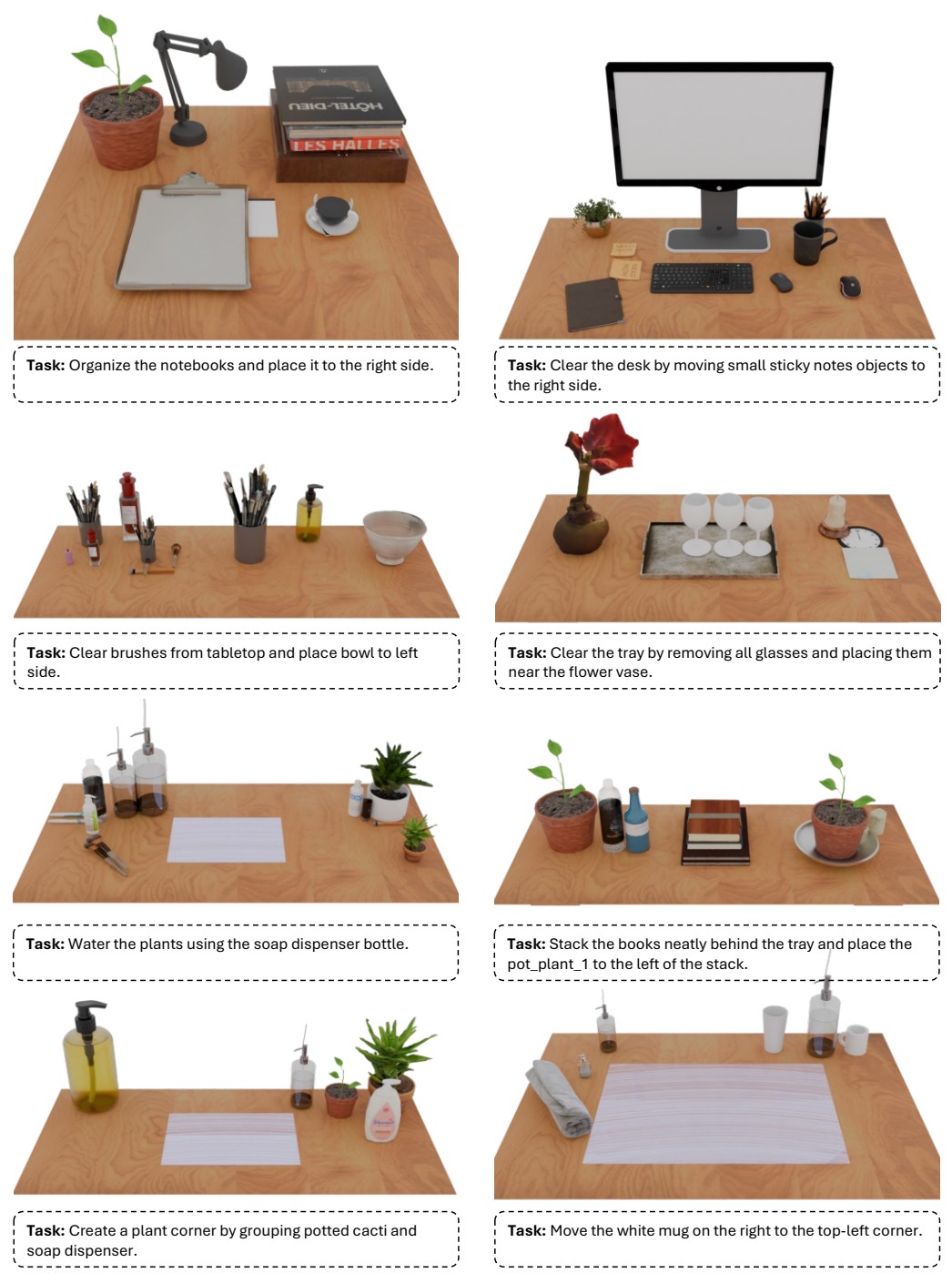

**Task:** Organize the notebooks and place it to the right side.

**Task:** Clear the desk by moving small sticky notes objects to the right side.

**Task:** Clear brushes from tabletop and place bowl to left side.

**Task:** Clear the tray by removing all glasses and placing them near the flower vase.

**Task:** Water the plants using the soap dispenser bottle.

**Task:** Stack the books neatly behind the tray and place the pot_plant_1 to the left of the stack.

**Task:** Create a plant corner by grouping potted cacti and soap dispenser.

**Task:** Move the white mug on the right to the top-left corner.

Figure 12: Additional qualitative results generated by our proposed MesaTask (continued).

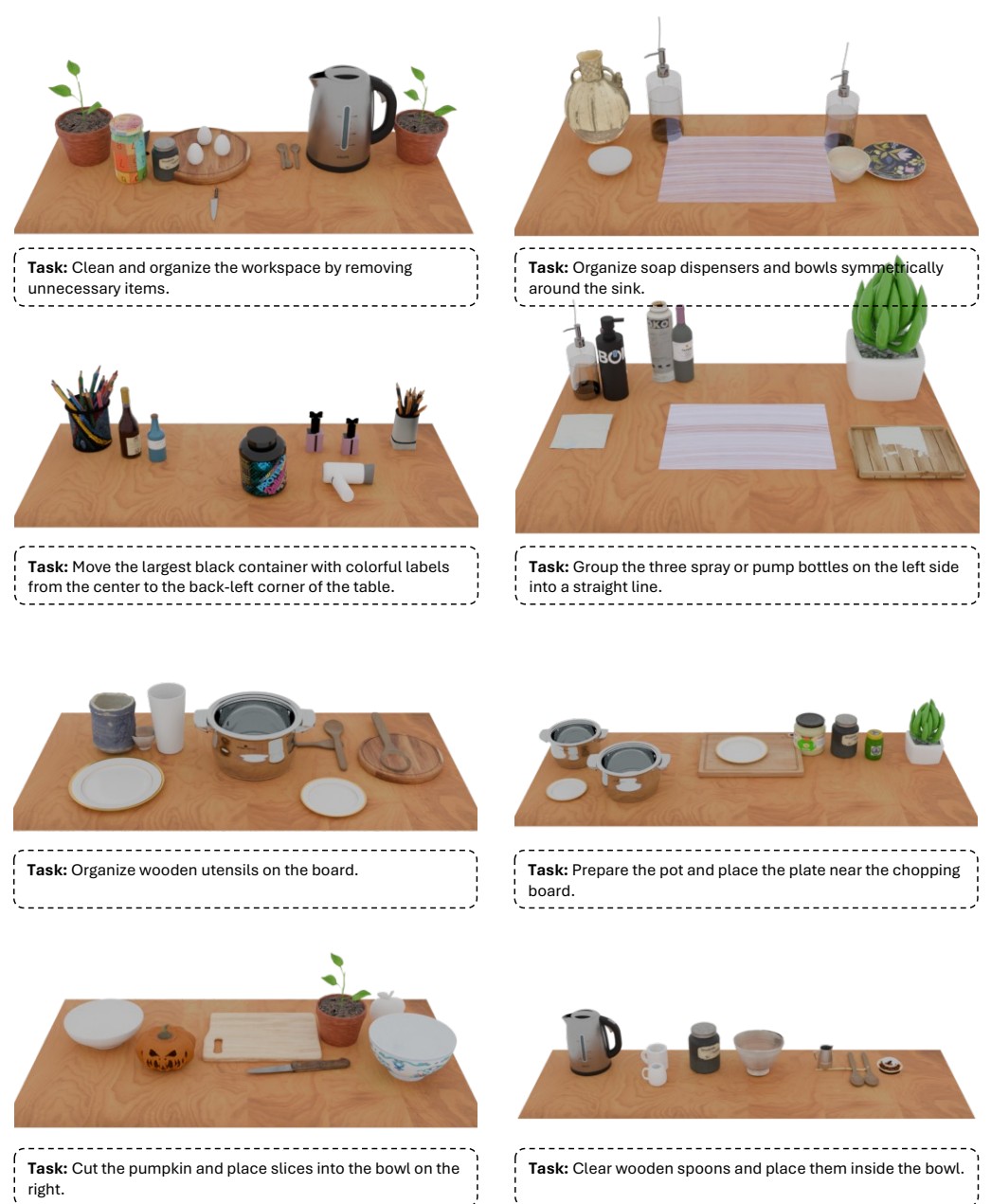

**Task:** Clean and organize the workspace by removing unnecessary items.

**Task:** Organize soap dispensers and bowls symmetrically around the sink.

**Task:** Move the largest black container with colorful labels from the center to the back-left corner of the table.

**Task:** Group the three spray or pump bottles on the left side into a straight line.

**Task:** Organize wooden utensils on the board.

**Task:** Prepare the pot and place the plate near the chopping board.

**Task:** Cut the pumpkin and place slices into the bowl on the right.

**Task:** Clear wooden spoons and place them inside the bowl.

Figure 13: Additional qualitative results generated by our proposed MesaTask (continued).

# E   Prompt

## E.1   Table Image Generation Prompt

**Table Image Generation Prompt:** Help me to generate a realistic table type: $tabletype$ placement description, this description will become my input to the picture to generate the model of the prompt, the requirements of a variety of items, a variety of relationships between the placement of a variety of reasonable, list the items that appear in the image, and then generate the brief location relationship.

Attention:
1. Do not generate items on walls or hang them on walls like a mirror or a picture.
2. Do not generate rare small items.
3. Do not generate chairs, only desktop placements.
4. Do not generate people, only items.
5. Do not generate tablecloths or table mats.
6. Generate images that show the entire tabletop, with the items displayed in their entirety.
7. Do not generate shelves or cabinets.
8. Only generate items can be placed on the table.
9. Give simple item names (1 or 2 words).
10. The number of Items should be $numitems$.
11. "COFFEE TABLE" does not necessarily have to have coffee. It is mainly a low table that is placed near the sofa in the living room.
Generate five prompts, each of which should be 80 words or fewer. You should only output the prompt; do not output any other content, list number, or true.
The output template should be: $tabletype$ parallel to the picture. blank wall, frontal top view, photograph. Items: .

## E.2   Inference Prompt

**Task Instruction to Task Info Prompt:** You are an AI assistant for robotic task planning. Given a high-level abstract task, expand it into a structured JSON format.
Input: A single string: "Task" (e.g., "High-level abstract task < 20 words").
Output Format (Strict JSON):
{
"Environment": "Brief scene description for the task",
"Task": "The original input task string",
"Goal": ["Ordered sub-objectives to achieve the task (aim for >=3 objects if task allows)"],
"Action Sequence": ["Primitive robotic actions with parameters (e.g., Pick(Object))"],
"Objects cluster": ["List of unique object types involved"]
}
Instructions for Fields:
1. Task: Copy the input task string directly.
2. Environment: Briefly describe a plausible environment for the task.
3. Goal: Break down the task into ordered sub-objectives. Involve at least 3 object types if the task context permits; otherwise, use only necessary objects.
4. Action Sequence: List primitive actions for the goals. Use object *types*. * Available actions: 'Pick(obj)', 'PlaceOn(obj)', 'PlaceAt(pos)', 'Push(obj, dir, dist)', 'RevoluteJointOpen(obj)', 'RevoluteJointClose(obj)', 'PrismaticJointOpen(obj)', 'PrismaticJointClose(obj)', 'Press(obj)'.
5. Objects cluster: List all unique object types from the task, goals, and actions.
Ensure the output is only the JSON object.

### E.3 GPT-score Evaluation Prompt

**System Prompt** You are an expert evaluator for 3D desktop scene layouts. Your task is to analyze desktop scenes and provide a detailed assessment based on specific criteria.
Please carefully examine the provided front and perspective views of the desktop scene, along with the task description.
Analyze how well the scene layout aligns with the intended task. Consider both the visible objects and their arrangement in relation to the task requirements.
You must provide numerical scores (1-10) for each criterion along with brief explanations to justify your ratings.
Be objective and consistent in your evaluation across different scenes.

**User Prompt** You are an expert at evaluating desktop scene layouts. Given both a front and a perspective view of a desktop scene, and a description of the tasks that can be performed on the desktop, please rate the scene's quality on a scale from 1 (poor) to 10 (excellent) according to the following criteria. For each criterion, consider both views:

1. **Consistency with Task:** Does the scene layout (objects present, their arrangement and relevance) align well with the provided task description (environment, objects, goals)?
- High score (7-10): All key objects are present, their arrangement is entirely logical and directly contributes to the task. The scene perfectly reflects the task requirements.
- Mid score (4-6): Most key objects are present and generally align with the task, but there might be minor inconsistencies or some less relevant elements.
- Low score (1-3): Significant deviations from the task description. Important objects are missing, or the arrangement is largely unrelated or illogical for the task.

2. **Object Size Reasonableness:** Are the sizes of the objects in the scene highly realistic, both relative to each other and to the overall desktop environment? Are they consistent across all objects?
- High score (7-10): All objects have highly realistic and perfectly proportionate sizes. No inconsistencies are noticeable.
- Mid score (4-6): Most objects have reasonable sizes, but there might be slight or occasional inaccuracies in proportion for some items.
- Low score (1-3): Multiple or glaring inconsistencies in object sizes. Some objects are obviously and significantly too large or too small, severely impacting realism.

3. **Placement Plausibility & Intersections:** Are objects placed stably and naturally on surfaces (e.g., not floating)? Are there any signs of unnatural intersection or penetration between objects? Objects should appear physically grounded and interact realistically.
- High score (7-10): All objects rest naturally and stably on surfaces. No aphysical interactions or intersections are visible. Objects are well-supported.
- Mid score (4-6): Most objects are placed plausibly, but there might be minor, subtle issues like slight floating or minimal, non-critical intersections.
- Low score (1-3): Obvious or frequent issues with object placement. Objects float, unnaturally overlap, or clearly penetrate each other, indicating a lack of physical realism.

4. **Layout Coherence & Realism:** Does the overall arrangement look highly functional, convincingly realistic, and typical for the task context? Does it avoid being overly staged, unnaturally sparse, or chaotically cluttered?
- High score (7-10): Layout is highly functional, convincingly realistic, and well-suited for the described task. The scene feels authentic and natural.
- Mid score (4-6): The layout is generally coherent and functional, but might lack some fine-tuning for optimal realism or could appear somewhat staged.
- Low score (1-3): Layout is chaotic, illogical, too empty, overly cluttered, or looks clearly artificial and unrealistic for the task.

5. **Object Visibility:** Are important objects mentioned in the task easily and unambiguously identifiable in at least one of the views? Are they sufficiently well-lit and resolved?
- High score (7-10): All key objects are clearly and unambiguously visible and identifiable. Their details are well-resolved.
- Mid score (4-6): Most important objects are visible, but some might require closer inspection to identify.
- Low score (1-3): Critical objects are very difficult to identify, or completely missing from view, hindering task understanding.

**Task Description:**

{task_description}

**Please provide a single score (1-10) for each criterion.**
**Strictly output in the following JSON format with no additional text:**

```
{
  "Evaluation": [
    {
      "criterion": "Consistency with Task",
      "explanation": "Your detailed explanation here",
      "score": X
    },
    {
      "criterion": "Object Size Reasonableness",
      "explanation": "Your detailed explanation here",
      "score": X
    },
    {
      "criterion": "Placement Plausibility & Intersections",
      "explanation": "Your detailed explanation here",
      "score": X
    },
    {
      "criterion": "Layout Coherence & Realism",
      "explanation": "Your detailed explanation here",
      "score": X
    },
    {
      "criterion": "Object Visibility",
      "explanation": "Your detailed explanation here",
      "score": X
    }
  ]
}
```

**Where X is an integer score from 1 to 10. Do not output anything else.**

### E.4 Holodeck-Table Prompt

**Table Prompt:** You are an experienced indoor designer, focusing on optimizing space and arranging objects on the tabletop appropriately. Please assist me in crafting a tabletop. Each table is a rectangle. You need to define the four coordinates. Note: the units for the coordinates are meters. For example:

- Computer Desk: [(0, 0), (0, 0.8), (0.5, 0.8), (0.5, 0)]
- Writing Desk: [(0, 0), (0, 0.6), (0.4, 0.6), (0.4, 0)]

Here are some guidelines:

1. A table's size ranges from 0.5m to 2m in length or width. The maximum area is 4 $m^2$. Provide a tabletop within this range.

2. Desktop types include, but are not limited to, computer desk, writing desk, dining table, coffee table, study desk, and bar table.

Now, I need a design for {input}. Additional requirements: {additional_requirements}. Your response should be direct and without additional text at the beginning or end.

---

**Object Selection Prompt:** You are an experienced indoor designer, focusing on optimizing space and arranging objects on the tabletop appropriately. Please assist me in selecting objects to decorate the table. Provide a description and desired size for each object in JSON format:

```
{
    "object_name": {
        "description": "A short sentence describing the object.",
        "size": [length, width, height],
        "quantity": number,
        "variance_type": "same" or "varied"
    },
    ...
}
```

For example:

```
{
    "Flower Vase": {
        "description": "A clear glass vase filled with fresh flowers.",
        "size": [10, 10, 20],
        "quantity": 1,
        "variance_type": "same"
    },
    ...
}
```

Currently, we are working on the {TABLE_TYPE} with a size of {TABLE_SIZE}. Please also consider the following additional requirements: {additional_requirements}.

---

**Object Constraints Prompt:** You are an experienced indoor designer, focusing on optimizing space and arranging objects on the tabletop appropriately. Help me arrange objects on the tabletop by assigning constraints to each object:

1. Global constraint:
    - edge: at the edge of the table.
    - middle: not close to the edge of the table.
2. Distance constraint:
    - near, object: near to another object, distance < 15cm.
    - far, object: far away from another object, distance >= 15cm.
3. Position constraint:
    - in front of, object: in front of another object.
    - around, object: around another object.
    - side of, object: on the side (left or right) of another object.
    - left of, object: to the left of another object.
    - right of, object: to the right of another object.
4. Alignment constraint:
    - center aligned, object: align the center of the object with the center of another object.
5. Rotation constraint:
    - face to, object: face towards the center of another object.

For each object, provide one global constraint and select from the other constraint types to ensure an optimal arrangement. Format each constraint as: `object | global constraint | constraint 1 | constraint 2 | ...`
Here are some guidelines:

- Start with an anchor object that does not depend on other objects.
- Place larger objects first and ensure objects of the same type are aligned.

Now, design {table_type} with a table size of {table_size}. Here are the objects I want to place on the {table_type}: {objects}
Please explain your high-level design strategy first, then strictly follow the desired format for providing constraints (do not add any additional text at the beginning or end).

### E.5 I-Design-Table Prompt

**Initiate Prompt:** The table has the size `[table length]`m x `[table width]`m x `[table height]`m
User Preference (in triple backquotes): `[user input]`
Table layout elements on the table (in triple backquotes): `['north_edge', 'south_edge', 'west_edge', 'east_edge', 'middle_of_table']`

**Refiner Prompt:**
Context: We are refining the placement of a cluster of objects relative to each other. These objects are all children of a parent object. Parent Object ID: `[parent_id]` Children Object IDs in this cluster: `[obj_names]` The children objects are all positioned '`[prep]`' the parent object '`[parent_id]`'.
`[Insert possibilities string]`
Task: Define the spatial relationships (e.g., "left of", "right of", "in front of", "behind") BETWEEN the children's objects listed above. The goal is to arrange them neatly and logically within their shared space relative to the parent.

**Output Format:**
Your response MUST be a JSON object (presented plainly, without delimiters). The JSON object must have a single key `"children_objects"`, which is a list. Each item in the `"children_objects"` list must be a dictionary representing one of the children.
Each child's dictionary must contain:

- `"name_id"`: the ID of the child object itself (from the list).
- `"placement"`: a dictionary describing placement relative to **other children** in the same cluster:
    - `"children_objects"`: a list of dictionaries, each defining a relationship:
        * `"name_id"`: the ID of the other child.
        * `"preposition"`: the spatial preposition (e.g., `"left of"`, `"right of"`).
        * `"is_adjacent"`: `true` or `false`, whether they are adjacent.

**Example:**

```
{
  "children_objects": [
    {
      "name_id": "apple",
      "placement": {
        "children_objects": [
          {
            "name_id": "banana",
            "preposition": "left of",
            "is_adjacent": true
          }
        ]
      }
    }
  ]
}
```

**Layout Refiner:** Every time the Admin speaks, look at the parent object (e.g., a table or a book) and its children objects (e.g., cup, pen, phone). Identify the first preposition connecting them, and then suggest a second relative positioning among the children. Use the JSON schema below:

```json
{
  "children_objects": {
    "type": "array",
    "items": {
      "type": "object",
      "properties": {
        "name_id": {
          "type": "string"
        },
        "placement": {
          "type": "object",
          "properties": {
            "children_objects": {
              "type": "array",
              "items": {
                "type": "object",
                "properties": {
                  "name_id": {
                    "type": "string",
                    "description": "The name_id of the other child object
                    "
                  },
                  "preposition": {
                    "type": "string",
                    "description": "e.g. left of the cup, behind the
                    phone",
                    "enum": ["on", "left of", "right of", "in front",
                    "behind", "under", "above"]
                  },
                  "is_adjacent": {
                    "type": "boolean",
                    "description": "Whether the objects are adjacent"
                  }
                },
                "required": ["name_id", "preposition", "is_adjacent"]
              }
            }
          },
          "required": ["children_objects"]
        }
      },
      "required": ["name_id", "placement"]
    }
  }
}
```

## E.6 Reasoning Data Construction Prompt

**Task from Scene Graph:** As an embodied task planner, analyze the given scene graph to generate diverse robotic manipulation tasks. Follow these guidelines:
Task Requirements:

- Propose one high-level tasks combining primitive actions
- Assume you are a human, you can command the assistant to do the task
- The proposed tasks should be as diverse as possible
- Each task must involve at least **3** objects with spatial reasoning. (The number of objects should be as many as possible, but if objects in scene graph are less than 3, only use the objects in scene graph)
- Use object types (e.g., "Dinner Plates") not instance IDs (e.g., "Dinner Plates-0")
- Consider object states (open/closed, filled/empty, etc.) and spatial relationships (near/far, left/right)
- Consider object affordance (e.g., container, cut, etc.)
- If there is a "in" or "above" relation, consider design the sub-goal to take the object out of the container or from the top of the object.
- Consider the complicated action, like put the object on the top of the object, or put the object in the container.

Action Constraints: You should consider the primitive actions the robot arm could take:

- Pick(obj_name)
- PlaceOn(obj_name)
- PlaceAt(position)
- Push(obj_name)
- RevoluteJointOpen(obj_name)
- RevoluteJointClose(obj_name)
- PrismaticJointOpen(obj_name)
- PrismaticJointClose(obj_name)
- Press(obj_name)

Output Structure: For each task, provide results:

```
{
    "Environment": "Brief scene description",
    "Task": "High-level abstract task (less than 20 words)",
    "Goal": [
        "Ordered sequence of sub-objectives",
    ],
    "Action Sequence": [
        "Primitive actions with parameters (e.g., Pick(Dinner Plate))"
    ],
    "Objects cluster": ["List of involved object types"]
}
```

The two given images are the front view and perspective view of the tabletop (corresponding to the scene graph), you can use the images to help you generate reasonable tasks.
The given tabletop description can be used to help you understand the scene and generate more reasonable and diverse tasks.

**Table Description:** The given images are the front view (in which the table is symmetric) and the perspective view of a tabletop. Only describe the layout on the tabletop; do not include the description of the table.

Please describe the scene in detail, including the objects (**do not describe the color/material/color of the object**), their positions, and the overall layout of the desktop, including empty and densely packed areas Output the description in a paragraph, no more than 200 words.

**Reasoning Context Generation:** Objective: Generate a reasoning process explaining the initial object arrangement for the task. Start by identifying objects, inferring context, and then providing placement reasoning as a paragraph, concluding with a transition to the scene graph. Ensure consistency with the reference `scene_graph` without explicitly mentioning the comparison.

Input:

1. `task_goal_object`: Describes the task, goals, actions, and core objects.

2. `scene_graph`: Provides the ground truth layout *for internal consistency reference only*.

Output: Structure the output as follows:

1. Introductory Sentence: Briefly state the goal is to determine the setup based on the task.

2. Scene Context Inference: Briefly infer the environment type based on the objects and task described in `task_goal_object` (e.g., "The task suggests a typical office desk setting.").

3. Core Task Objects: Use the exact header `Core Task Objects:` followed by a list of object types and counts (no instance names, derived from `task_goal_object["Objects"]`).

4. Environment Objects: Use the exact header `Environment Objects:` followed by a list of other object types and counts present in the context (no instance names, derived from `scene_graph`). Their presence should align with the inferred scene context. Avoid explicitly mentioning verification against the scene graph.

5. Placement Reasoning Paragraph: A coherent paragraph of detailed scene description, including the objects, their positions, and the overall layout of the tabletop inferred from the given `environment/Action Sequence/Goal` in task info. Consider accessibility, non-interference, and the *overall plausible layout*. Use the input reference `scene_graph`, `input scene images`, `scene description` *internally* to ensure resulting locations mentioned (e.g., 'middle_left') are correct, but do NOT state this comparison explicitly, and do NOT simply list relationships from `scene_graph`. Reference to the input scene description and scene images to make the paragraph more detailed and accurate.

6. Concluding Transition Sentence: End the paragraph with a transition sentence leading into the scene graph description (e.g. "Based on this task analysis, the scene graph is arranged as follows:"). Avoid generic evaluative summaries.

— Input Data —

Task/Goal/Object Description:

{task_goal_object}

