# OpenReview forum: "MesaTask: Towards Task-Driven Tabletop Scene Generation via 3D Spatial Reasoning"
_NeurIPS.cc/2025/Conference — NeurIPS 2025 spotlight_

### Official Review · Reviewer_WdWW · 2025-06-19

**Clarity:** 2
**Significance:** 3
**Originality:** 3
**Rating:** 4
**Confidence:** 5

**Summary:**

This paper presents 1) MesaTask-10K, a large-scale dataset of synthetic tabletop scenes, derived from image generation, automatic coarse scene construction and human refinement, 2) an LLM-based framework for scene generation that utilizes spatial reasoning chain, which decomposes the generation process into object inference, spatial interrelation reasoning, and scene graph construction, trained with SFT and DPO algorithms. Experiments are conducted on the MesaTask-10k dataset, compared with zero-shot modular scene generation methods, with user study and ablation on DPO training.

**Questions:**

I will consider raising my rating if the author could address my concerns over the dataset construction and evaluation in the weakness section, including the following

- What was the level of effort required from human annotators—were their contributions marginal or extensive? Please provide concrete details, including annotation procedures, task complexity, and average annotation time per sample. I would assume the results from the coarse construction stage bear severe implausibility.

- How to ensure that all scene layouts derived from image generation are physically and semantically plausible?

- How to unify the rotation angle around the axes for all the object assets, given that Objaverse assets lack standardized orientation metadata? This is critical for the object retrieval and placement in the synthesis process.

- With a library reportedly containing over 12,000 rigid and articulated 3D assets, how are these categories distributed?

- How to get the scale of the scene from DepthAnything v2?

**Ethical Concerns:**

["NO or VERY MINOR ethics concerns only"]

**Final Justification:**

After reading the author's rebuttal and opinions from other reviewers, I believe critical aspects, e.g., the annotation details, the metric scale of the scene, and physical metrics, should be incorporated in the revision. I believe the introduced dataset will be beneficial for the community if they are fully open-sourced with easy integration with BLENDER and Isaac. Thus, I raise my score accordingly.

**Limitations:**

yes

**Paper Formatting Concerns:**

The authors should use the correct citation style.

**Quality:**

2

**Strengths And Weaknesses:**

[+] S1: The paper addresses an important problem in scene synthesis, specifically focusing on tabletop scenarios. This is characterized by a higher diversity of object categories and complex spatial relationships, such as stacking and containment, which are particularly relevant for embodied manipulation tasks.

[+] S2: The proposed method decomposes the scene generation into distinct stages: object inference, spatial interrelation reasoning, and scene graph construction. This modular design is logically coherent and amenable to integration with chain-of-thought (CoT) style LLM-based architectures.

[+] S3: Qualitative results in both the main paper and supplementary material suggest that the method generates plausible tabletop scenes in response to task-specific prompts.

[-] W1: While the dataset has the potential to be a substantial contribution to the community and the related field, the paper currently lacks sufficient detail and justification regarding its quality. Critical missing aspects include: the diversity and quality of image generation, the quality and consistency of coarse scene construction, the coverage and alignment of 3D assets corresponding to objects in the generated images, the detailed labor and methodology of human annotation, and the overall visualization and quality control of the dataset. The dataset quality also justifies the validity of the proposed method. Providing extra details of the dataset construction—such as figures, representative samples, initialization, and annotation workflows—would significantly enhance clarity.

[-] The main paper only reports comparisons with zero-shot baselines. While quantitative results for methods like ATISS, DiffuScene, and PhyScene are included in the supplementary material, no direct comparison with the proposed method is made. Additionally, the evaluation lacks metrics assessing physical plausibility, which is articulated as the motivation and distinct characteristic of the paper.

*Minor*

- L35-L38 unclear sentences with grammar errors

- Typo L166-L169 Figure 3 should be Figure 2

---

> ### Author Rebuttal · Authors · 2025-07-31
>
> We thank the reviewer for the encouraging feedback and incisive questions. Below we address your questions and weaknesses mentioned:
>
> ## The diversity/quality of image generation.
>
> We ensure the diversity of generated images through two steps. First, the LLM is capable of producing diverse scene descriptions, encompassing a rich object categories and relationships. Second, as an SOTA text-to-image model, FLUX generates images that are both diverse and rich in content.
>
> In terms of parameter setting, we assign distinct random seeds to each image to guarantee the randomness and diversity of generated content. We set the guidance_scale to 3.5, a relatively low value. This balances prompt consistency with greater creative flexibility for the model, thereby enhancing the diversity. Additionally, we set the output image resolution to 1024×1024 to ensure overall visual quality.
>
> ## The quality and consistency of coarse scene construction.
>
> The quality of the generated 3D scenes is poor. Issues such as incorrect sizes, collisions, and redundant objects frequently occur, as mentioned in Figure 2 of the main paper. These problems arise because our pipeline produces numerous redundant boxes. For instance, in Figure 2, a lemon, a bowl, and the combination of the lemon and bowl each have their own bounding boxes.
>
> To ensure the consistency between the 3D scene and the corresponding image, we do not filter out redundant bounding boxes. This decision is based on two considerations:
>
> 1. Rule-based filtering might inadvertently remove bounding boxes of objects that should be included, thereby compromising consistency.
>
> 2. Following coarse scene construction, annotators perform manual crafting of the scenes. At this stage, removing redundant objects requires only simple operations.
>
> ## The coverage and alignment of 3D assets corresponding to objects in the generated images.
>
> Our large-scale 3D asset database covers most objects present in the scenes, though some objects in the generated images may not be included. To address this, we first record objects with a retrieval similarity score below 0.7 during 3D asset retrieval, indicating they are not covered by the database. After manual inspection, we generate corresponding 3D assets using a 3D generation model (Hunyuan3D) and add them to the asset database. During the layout refinement phase, annotators report objects present in the images but missing from the coarse 3D scenes; we expand the 3D asset library through the same process. In total, 222 generated 3D assets are additionally included in our asset library.
>
> ## The detailed labor and methodology of human annotation.
>
> During annotation, annotators are provided with the coarse 3D scene in GLB format, which includes a Unitree H1 robot model (with an absolute height of 1.7m) to facilitate the construction of metric-scale 3D scenes, along with reference images and all object snapshots from the images.
>
> Annotators import the GLB file into Blender to refine the scene. They adjust each object’s relative size and position with reference to the given tabletop scene images, calibrate the overall scene scale using the H1 model, and rotate objects to match their orientations in the images.
>
> ## The overall visualization and quality control of the dataset.
>
> Quality control of the dataset. After receiving each annotated 3D file from annotators, we render all annotated 3D scenes from four directions: front, back, left, and right. These renderings are compared with reference images. If quality fails to meet requirements, such as frequent issues like unreasonable object sizes or objects floating in the air, we request annotators to revise the errors until they meet the acceptance criteria.
>
> Dataset overall visualization. We have shown some tabletop scenes in MesaTask-10k in Figure 1 of the main paper. Besides, we will open-source the MesaTask-10K dataset upon paper acceptance. The dataset will include four-view renderings of each scene.
>
> ## Comparison with ATISS, DiffuScene, and PhyScene.
>
> Different settings. Our approach focuses on generating scenes from task instructions, whereas prior methods such as ATISS, DiffuScene, and PhyScene generate scenes from simple scene descriptions. This fundamental difference in settings precludes direct comparisons.
>
> Open-vocabulary task-to-scene generation capability. Our method enables open-vocabulary task-to-scene generation. Leveraging a fine-tuned LLM to generate scene layouts, it can produce layouts and textual descriptions for objects not present in the training set. In contrast, ATISS, DiffuScene, and PhyScene generate predefined object features and cannot generate objects out of the training set at inference time.
>
> ## Physical plausibility metrics.
>
> Thanks for the suggestion. In tabletop scenes, there exist numerous complex interrelations, such as stacking and containment. To calculate physical plausibility metrics, we thus evaluate occupancy overlaps between objects rather than bounding box intersections.
>
> Specifically, the layout of each scene, along with corresponding object assets, is transformed into Drake’s[1] internal scene tensor representation. We then use Drake’s geometry engine to compute signed distances between all pairs of collision geometries. A negative signed distance indicates interpenetration, which is counted as a collision event.
>
> We obtained physical plausibility results on the test set in the main paper, with the collision rate metric defined as the number of collision object pairs $N_{collision}$ divided by the total number of potentially collision object pairs $N_{total}$. As shown in the table below, MesaTask generates collision-free scenes after physics-based post-processing, satisfying physical plausibility.
>
> | Method                                 | GPT-4o | I-Design-table | Holodeck-table | MesaTask w/o physical simulation | MesaTask |
> |----------------------------------------|--------|----------------|----------------|----------------------------------|----------|
> | Collision Rate (%)                     | 21.13  | 9.92           | **0**          | 11.21                            | **0**    |
>
>
> Thanks for pointing out the problem. We will revise them in the camera-ready version.
>
> ## What was the level of effort required from human annotators?
>
> We agree that the coarse construction stage is not sufficiently plausible due to challenges such as occlusion, inaccurate depth estimation, and retrieval errors. Thus, human annotators played a critical and extensive role in refining the layout.
>
> Specifically, for each sample, annotators were provided with:
> - The GLB file of the tabletop scene,
> - The rendered scene image,
> - Individual object snapshots and their indices.
>
> Using Blender, annotators manually adjusted the scene layout to match the reference image. This included:
> - Translating objects to correct positions,
> - Scaling them to appropriate sizes,
> - Rotating them to align orientations correctly,
> - Ensuring spatial plausibility of inter-object relations.
>
> The annotation complexity varied substantially depending on the number of objects and the complexity of their spatial configurations. For example, scenes with many small objects and dense relations were significantly more complex and time-consuming. On average, annotators spend 10 to 20 minutes per scene.
>
> ## Physically and semantically plausibility of generated images.
>
> During the image generation phase, we note that images generated by models trained on large-scale image-text data are inherently physically and semantically plausible. Even if an image has issues, annotators construct scenes based on reference images and ensure the physical and semantic plausibility of scene layouts using common sense.
>
> In the review phase after receiving annotations, we inspect each sample. If a layout fails to meet physical or semantic plausibility criteria, it is returned for re-annotation. Given the large number of small objects in tabletop scenes, where collisions or overlaps might be overlooked, we also run the annotated scenes through a simulation environment for physical simulation to ensure physical plausibility.
>
> ## Standardized orientation of object assets.
>
> Our 3D asset database includes objects curated from Holodeck, the PartNet Mobility dataset, and assets generated by the image-to-3D model (Hunyuan3D). Objects in PartNet-Mobility are already orientation-aligned.
> For the other 3D assets, we first centered the camera around the object and uniformly rendered eight views around the z-axis (up). We manually exclude assets that cannot be standardized via z-axis rotation. A VLM model then identifies which of the eight images shows the front view, and the 3D asset is rotated accordingly.
> After rotating the assets, we manually inspect each object’s orientation. Assets not properly rotated to face the front are re-oriented manually. This annotation pipeline ensures all objects in our database are orientation-aligned.
>
> ## Category distribution of assets in the database.
>
> Our 3D asset library comprises 11,000 rigid 3D assets, with the distribution of the top 100 object categories presented in Figure 1 of the supplementary material. It also includes 1,034 articulated objects, selected from PartNet Mobility and belonging to 26 tabletop object categories.
>
> ## The scale of scenes.
>
> DepthAnything v2 provides only relative depth information of objects, which is insufficient for downstream embodied simulation. To obtain the real scene scale, we include a Unitree H1 robot with metric size beside the table. Annotators then scale objects in the scene with reference to the robot’s size.
>
> During the review of annotations, we check the relative proportions between objects in the scene and the H1 robot model to assess annotation quality. If the proportions are incorrect, the scene will be re-annotated.
>
> [1] R. Tedrake and the Drake Development Team. Drake: Model-based design and verification for robotics. 2019.

---

> > ### Comment · Reviewer_WdWW · 2025-08-04
> >
> > I thank the authors for their detailed response. Several critical aspects, which are currently missing in the paper, should be incorporated in the revision:
> > - The level of effort required from human annotators: 10-20 minutes per scene requires significant time for the annotation. This should be considered as a major effort (as well as a contribution) towards a high-quality dataset. Rather, the current framing understates this by emphasizing a “semi-automatic pipeline with human assistance.” I believe the paper should be faithful on this point.
> > - The scale of scenes: The inclusion of the Unitree H1 robot model is missing and should be documented.
> > - Physical plausibility metrics
> >
> > That said, I believe the paper and the introduced dataset will be beneficial for the community, provided they are fully open-sourced and offer easy integration with BLENDER and Isaac.

---

> > > ### Author Response · Authors · 2025-08-04
> > >
> > > Thanks for your feedback. We will take your suggestions and incorporate these aspects in the revision. The models and datasets will be fully open-sourced, and we will ensure that the 3D scenes are fully compatible with BLENDER and Isaac.

---

### Official Review · Reviewer_HbVw · 2025-07-01

**Clarity:** 4
**Significance:** 4
**Originality:** 4
**Rating:** 5
**Confidence:** 5

**Summary:**

This paper presents a novel task, task-oriented tabletop scene generation, which aims to generate realistic tabletop scenes from high-level task instructions. They also curate a large-scale MesaTask-10K dataset, which contains rich spatial relations. They propose a chain-of-though (CoT) method, which adopts an LLM to derive intermediate processes for the final layout.

**Questions:**

+ When constructing MesaTask-10K, the LLM is first employed to derive kitchen descriptions. How diverse are the generated scenes? Is there a way to explicitly encourage the LLM to produce diverse content, or is just adjusting the temperature during inference?
+ Again, they only adopt Flux.1 as the image generation model. Since all the descriptions pertain to kitchen tabletops, are the generated images sufficiently diverse? Moreover, the LLM output is expected to be a dense description—can Flux.1 handle such lengthy input effectively? What are the recall rates for the mentioned objects and spatial relations in the generated images?
+ "During the Spatial Reasoning Chain, the entire process depends on each intermediate step produced by the LLM. However, these steps are not guaranteed to be perfect. There should be an analysis of the accuracy of each step, how errors propagate through subsequent steps, and how they affect the final outcome. Such an analysis would offer valuable insights into the reasoning chain and help identify opportunities to improve each step if necessary.

**Ethical Concerns:**

["NO or VERY MINOR ethics concerns only"]

**Limitations:**

yes

**Quality:**

3

**Strengths And Weaknesses:**

**Main Strengths**
+ This paper is well-written and easy to follow.
+ The proposed task is novel and practical, with significant potential for real-world applications, and can extend to various indoor and outdoor design scenarios.
+ The authors provide lots of qualitative examples in the Appendix, including intermediate information generated by the LLM, which is helpful for case studies and future research.

**Main Weaknesses**
+ One of the most valuable contributions of this paper is the curated MesaTask-10K dataset. However, I could not find a shareable link in the draft. If the authors do not plan to open-source it, this may reduce the impact of the paper.

---

> ### Author Rebuttal · Authors · 2025-07-31
>
> We thank the reviewer for the encouraging feedback and incisive questions. Below we address your questions and weaknesses mentioned:
>
> ## Dataset open-source.
>
> We appreciate the recognition of our proposed MesaTask-10K. We plan to fully open-source the dataset upon acceptance of the paper, including the 3D asset library, full dataset visualization, the layout corresponding to each scene, and the code for constructing a 3D scene based on the layouts. We hope this will contribute to advancing the community and facilitating real-world applications.
>
> ## Diversity of generated scene description.
>
> In Figure 2 of the main paper, we prompt the LLM to generate descriptions for six common indoor tabletop scenes, including Coffee Table, Dining Table, Dressing Table, Kitchen Counter, Office Table, and Bathroom Vanity, **among which Kitchen Counter is one specific category**. This diversity across tabletop types ensures the richness of object types and layouts.
>
> ## Explicit ways to generate diverse content.
>
> We rely on well-designed prompts to encourage the LLM to generate diverse tabletop scene descriptions, without employing other methods such as adjusting the temperature. This is because, during dataset construction, object sets and arrangements in each scene description are distinct. Consequently, tabletop scene descriptions exhibit significant diversity. The specific LLM prompts are provided in Section F.1 of the supplementary materials.
>
> ## Diversity of generated images.
>
> In the construction of the MesaTask-10K dataset, we generate descriptions for six common tabletop scene categories, with the kitchen counter being one of them. When using Flux.1 for image generation, we assign distinct random seeds to each image to ensure the randomness and diversity of generated content. Additionally, we set the guidance_scale to 3.5. This relatively low value maintains consistency with prompts and also enhances image diversity. These measures collectively ensure that the generated images are sufficiently diverse.
>
> ## Can Flux.1 handle dense scene descriptions?
>
> We constrain text length during LLM-based scene description generation, utilizing the instruction: "Generate five prompts, each limited to 80 words or fewer." Complete scene description generation prompts are provided in Section F.1 of the supplementary materials.
>
> Additionally, we configured Flux with a max_sequence_length of 512 for image generation. This configuration enables it to process text descriptions with a maximum length of 512 tokens.
>
> ## Alignment between text descriptions and generated images.
>
> Flux.1 is a SOTA open-source text-to-image model. It has robust instruction-following capabilities. Flux.1 generally generates images that align with input descriptions.
>
> Notably, we prioritize layout alignment between 3D scenes and images over text-image alignment, as the diversity of layouts inherent in the images ensures the curation of diverse 3D tabletop scene layouts. Hence, there is no stringent requirement for the generated images to explicitly incorporate all objects and spatial relations specified in the descriptions.
>
> ## Analysis of the accuracy of each step in the Spatial Reasoning Chain.
>
> The Spatial Reasoning Chain consists of three steps: Object Completion, Interrelation Inference, and Scene Graph Generation.
>
> Thanks to the powerful capabilities of the LLM, we empirically find no errors in the Object Completion and Interrelation Inference steps.
>
> However, in Scene Graph Generation, there is a low probability that the generated scene graph may omit task-irrelevant objects mentioned in the Object Completion step. This results in the absence of certain task-irrelevant objects in the final layout. To mitigate this issue, an LLM (e.g., GPT-4o) can be employed to verify whether MesaTask outputs omit task-irrelevant objects referenced in the Object Completion step, and subsequently add layout information for these objects.

---

> ### Author Response · Authors · 2025-08-06
>
> Dear Reviewer,
>
> Thank you once again for reviewing our paper. We would greatly appreciate it if you could take a moment to review our feedback, and let us know if any concerns remain.
>
> Best regards,
>
> The Authors

---

### Official Review · Reviewer_2SRJ · 2025-07-02

**Clarity:** 3
**Significance:** 4
**Originality:** 4
**Rating:** 5
**Confidence:** 4

**Summary:**

This study introduces an innovative framework designed to enhance large language models (LLMs) with 3D spatial reasoning and tabletop scene generation capabilities. The framework employs a structured Spatial Reasoning Chain to decompose the task-to-scene generation process into sequential steps: object inference, spatial interrelation reasoning, and scene graph construction. Supervised fine-tuning (SFT) is used to train the model on high-quality, manually crafted layouts from the MesaTask-10K dataset. To address limitations such as object collisions or misaligned inter-object relations, the framework further integrates Direct Preference Optimization (DPO), which refines the model's outputs by learning from paired positive and negative layout examples. This dual approach ensures the generation of physically plausible and task-aligned tabletop scenes, as validated by extensive experiments and user studies.

**Questions:**

In addition to the above weaknesses, clarification on the following points are suggested:
1. How can the diversity of object categories generated by a large language model be effectively constrained to ensure task relevance and semantic consistency?
2. What systematic strategies can be employed to detect and correct structural issues, such as inaccurate bounding boxes or invalid spatial configurations, in generated layouts, and to what extent can Direct Preference Optimization (DPO) reliably address these issues at scale?

**Ethical Concerns:**

["NO or VERY MINOR ethics concerns only"]

**Final Justification:**

The authors have addressed most of the concerns to a satisfactory extent. Although the generalization capability could benefit from further improvement, the paper remains an insightful and commendable contribution. I therefore assign a final score of 5: Accept.

**Limitations:**

The quality and diversity of the generated tabletop scenes are constrained by the limitations of the existing 3D asset library. The current method is unable to generate novel objects beyond the predefined asset set, and it lacks the capability to model richer material attributes and object state variations.

**Paper Formatting Concerns:**

No major formatting issues are noticed.

**Quality:**

3

**Strengths And Weaknesses:**

Strengths
1. This work presents a comprehensive framework that equips large language models with spatial layout generation capabilities via reasoning-driven supervised fine-tuning (SFT), while leveraging Direct Preference Optimization (DPO) to correct suboptimal or semantically inconsistent outputs.
2. This work curated a high-quality tabletop layout dataset, which has the potential to catalyze future research efforts in this domain.
3. Conducting environment simulation within Isaac Gym facilitates seamless integration into sim-to-real pipelines for embodied intelligence

Weaknesses
1. The paper lacks a comparative analysis of the generated tabletop scenes across tasks of varying types and complexity levels, making it difficult to assess both the adaptability and the superiority of the proposed method in diverse task settings.
2. There is no in-depth discussion of representative success cases and failure modes from both the proposed approach and baselines, which hinders a thorough evaluation of practical performance and limitations.
3. The generalization capability has not been validated, and whether the proposed layout reasoning process can extend effectively to diverse scenarios beyond MesaTask-10K remains an open question.

---

> ### Author Rebuttal · Authors · 2025-07-31
>
> We thank the reviewer for the encouraging feedback and valuable comments. Below, we address your questions and weaknesses mentioned:
>
> ## Analysis of generated scene results across tasks of varying types and complexity levels.
>
> Thanks for your constructive suggestions. Following ManiTaskGen [1], we categorized tasks of varying types and complexity levels into four Task Difficulty Levels, specifically:
>
> - Level 1: Single-step pick-and-place tasks with a unique target object and no perceptual ambiguity (e.g., "Move the red dictionary on the bookshelf to the table").
>
> - Level 2: Single-step pick-and-place tasks with non-unique target objects, requiring additional descriptions for distinction (e.g., "Move the blue cup on the table to the coffee table" where multiple cups exist in the scene).
>
> - Level 3: Multi-step tasks formed by two Level 1 or Level 2 tasks connected by "THEN" (e.g., "First move the book from the bookshelf to the left of the table, then move it to the right of the table").
>
> - Level 4: Outcome-based abstract tasks describing the target scene state rather than specific steps (e.g., "Tidy up the messy desk", "Make the living room cleaner").
>
> We provided the definitions of these four task levels to GPT-4o to generate diverse tasks, yielding 500 tasks per level. Each level’s tasks cover six common indoor tabletops (bathroom vanity, dining table, kitchen counter, coffee table, dressing table, office table). We evaluated the generated scenes using the same metrics as in the main paper. As shown in the table below, tabletop scenes generated under all task levels achieved high scores in the multi-dimensional assessment, confirming that our method can effectively handle tasks of varying types and complexity levels.
>
> | Level   | Success Rate (%) | FID  | Consistency with Task (CwT)  | Object Size Reasonableness (OSR)  | Placement Plausibility & Intersections (PPI)  | Layout Coherence & Realism (LCR)  | Object Visibility (OV)  | Avg. |
> |---------|------------------|------|------|------|------|------|------|------|
> | level1  | 99.8             | 59.3 | 7.20 | 8.88 | 9.40 | 7.22 | 7.80 | 8.10 |
> | level2  | 99.4             | 55.5 | 7.44 | 8.40 | 9.36 | 7.44 | 8.16 | 8.16 |
> | level3  | 99.2             | 50.9 | 7.04 | 8.36 | 9.46 | 7.28 | 8.10 | 8.05 |
> | level4  | 98.4             | 43.9 | 7.46 | 8.78 | 9.70 | 7.68 | 8.88 | 8.50 |
>
>
> However, we observed variations in FID across different task levels. This discrepancy arises because task instructions in the training set are predominantly at Level 4 difficulty (when training set tasks were classified using GPT-4o according to the above criteria, 83.8% fell into Level 4, 11.4% into Level 3, 3.8% into Level 2, and 1% into Level 1). When open-sourcing the dataset, we will generate these four levels of task instructions uniformly for each scene to ensure comprehensive coverage of tasks with varying types and complexity levels.
>
> ## Analysis of success cases and failure modes from both the proposed approach and baselines.
>
> We analyze the success cases and failure modes of each method referenced in **Figure 4 of the main paper** and **Supplementary Figure 4**.
>
> The baselines commonly exhibit issues such as missing task-relevant objects, spatial collisions, illogical arrangements, and incorrect object orientations or groupings, particularly under dense layouts.
>
> Our proposed MesaTask produces scenes with **visually plausible layouts** and a **diverse range of objects**, as shown in all three examples in **Figure 4**. It also aligns well with task instructions, for example, books in the first scene are placed “mostly centered and to the back middle,” and the candle in the second scene appears on the tray as required. MesaTask further captures **fine-grained spatial relations** like stacking, above, and containment, highlighted in the zoom-in insets.
>
> In contrast, the baselines exhibit various failure modes. **GPT-4o** fails to generate valid 3D layouts, and objects are severely distorted (e.g., flattened books) and are placed in physically implausible positions. This is due to the lack of explicit spatial understanding in pretrained LLMs. **I-Design-table** retrieves more diverse assets but fails to produce correct object sizes, placements, or spatial relationships, as it similarly lacks grounded spatial reasoning. **Holodeck-table** performs better in reducing inter-object collisions but still suffers from inaccurate object scaling and lacks consistent alignment with the task goals.
>
> For failure cases in the MesaTask result. A typical example involves **open vs. narrow-mouthed bowls**, since both may share similar bounding box size, the model may place objects inside narrow bowls without considering the inward curvature, leading to initial overlaps. While physical simulation can resolve collisions, it may cause significant shifts in object positions. For instance, items inside the bowl may be ejected, thus resulting in physically valid but semantically incorrect layouts.
>
> ## Validation of generalization capability.
>
> To validate the generalization capability of our proposed method, we selected tabletop categories not present in MesaTask-10K, including nightstands, TV stands, and side tables from household scenes, as well as cashier counters from shop scenes. We used GPT4o to generate plausible tasks for these four tabletop categories, with 16 tasks per category. As shown in the table below, the results demonstrate strong performance across all metrics.
>
> | Category          | Success Rate (%) | CwT  | OSR  | PPI  | LCR  | OV   | Avg. |
> |-------------------|------------------|------|------|------|------|------|------|
> | Nightstand        | 100.0            | 7.44 | 8.12 | 9.00 | 7.69 | 8.75 | 8.20 |
> | TV stand          | 100.0            | 6.56 | 8.06 | 9.06 | 6.94 | 7.44 | 7.61 |
> | Side table        | 100.0            | 7.62 | 8.62 | 9.25 | 7.69 | 8.50 | 8.34 |
> | Cashier counter   | 100.0            | 6.25 | 8.38 | 9.06 | 6.62 | 7.50 | 7.56 |
>
> The results across all metrics are similar to the performance on the test set presented in Table 1 in the main paper. Notably, in the case of cashier counters, even though cash registers are not included in MesaTask-10K, our method can accurately generate their descriptions and sizes while placing them correctly.
>
> Besides, we did not compute FID for evaluation. For these four tabletop scenes contain numerous objects not present in MesaTask-10K, the generated scenes of these categories should not be similar to those in the dataset.
>
> ## How can the diversity of object categories generated by a large language model be effectively constrained to ensure task relevance and semantic consistency?
>
> We argue that constraining the object categories generated by LLMs is unnecessary in task-driven tabletop scene generation, as real-world tasks inherently involve a highly diverse range of objects. Instead, we propose establishing a systematic approach to detect object categories not covered by the 3D asset database and providing user-friendly tools for adding 3D assets, both of which effectively ensure task relevance and semantic consistency.
>
> In our method, after LLMs generate object information, we use the SBERT model to extract text features from the generated descriptions and compute their similarity with those of objects in the asset database. If the similarity falls below 0.7, the program issues a warning indicating that no suitable 3D model can be retrieved, along with a suggestion to expand the 3D asset library. We have also implemented a one-click script that supports batch addition of 3D assets in multiple formats (GLB/OBJ/URDF) to the database.
>
> It's noteworthy that our database contains over 12,000 tabletop objects, thus covering various object categories in most cases.
>
> ## Systematic strategies to detect and correct structural issues in generated layouts, and DPO's reliability in addressing them at scale.
>
> Inspired by the VLM-driven 3D asset annotation pipeline in Holodeck, we employ VLM to detect structural issues.
> As for the issues of inaccurate bounding boxes and invalid spatial configurations, we accordingly design two metrics: bounding box size correctness and Spatial Configuration Validity. First, we use GPT-4o to judge whether an object’s bounding box size is correct, and calculate the proportion of objects with correct bounding box sizes relative to the total number of objects. Additionally, we leverage GPT-4o to assess the validity of the scene graph within the spatial reasoning chain (e.g., unreasonable overlaps, absence of hierarchical conflicts, or logical positional contradictions), and calculate the proportion of scenes with a valid scene graph to the total number of test scenes.
>
> In our work, we construct negative samples by perturbing object sizes and removing key edges from the scene graph, then use DPO to perform post-training on the LLM. Experiments on the same test set as in the main paper (shown in the table below) demonstrate that DPO can mitigate these two issues to a certain extent.
>
> | Method | Bounding Box Size Correctness (%) | Spatial Configuration Validity (%) |
> |--------|-----------------------------------|------------------------------------|
> | SFT    | 94.38                             | 96.27                              |
> | DPO    | **95.14**                         | **96.96**                          |
>
>
> As for systematic strategies, we suggest employing an external VLM during test time to detect and correct these structural issues. According to the feedback from VLM, users can iteratively improve the results.
>
> [1] Dai L, Wang H, Wan W, et al. ManiTaskGen: A Comprehensive Task Generator for Benchmarking and Improving Vision-Language Agents on Embodied Decision-Making. (arXiv 2025)

---

> > ### Comment · Reviewer_2SRJ · 2025-08-06
> >
> > Thank the authors for their detailed response. The authors have resolved most of the major issues. However, several aspects still warrant further clarification.
> > 1. From an intuitive standpoint, it remains unlikely that a scene generation pipeline based on large language models can consistently achieve a 100% success rate, even when fine-tuned using Direct Preference Optimization (DPO). Challenges such as bounding box collisions and spatial inconsistencies are inherently difficult to eliminate. The authors themselves acknowledge in their response that errors may still occur in bounding box dimensions and spatial relationships. It would be helpful if the authors could clarify how this reported success rate is defined. Does it refer to the syntactic correctness of the LLM-generated output format, or are there additional mechanisms in place to validate and correct physical or spatial errors during generation?
> > 2. Regarding generalization, the authors present results on out-of-distribution tabletop scenes, which partially address concerns about the model’s ability to generalize beyond the training data. However, the original intent behind raising this issue was to explore whether the framework could extend to non-tabletop environments, such as full-room layouts involving furniture arrangement. It would be valuable if the authors could provide further discussion on this point. Specifically, insights into the model’s adaptability to broader indoor scene generation tasks would help clarify its potential applicability beyond the tabletop domain.

---

> ### Author Response · Authors · 2025-08-06
>
> Dear Reviewer,
>
> Thank you once again for reviewing our paper. We would greatly appreciate it if you could take a moment to review our feedback and newly added experiments, and let us know if any concerns remain.
>
> Best regards,
>
> The Authors

---

> ### Author Response · Authors · 2025-08-06
>
> Thank you for your constructive feedback, which helps us further clarify the key points. Below, we address your questions in detail:
>
> ## Defination of success rate.
> As clarified in the supplementary material, the reported success rate in our work refers to the syntactic correctness of the LLM-generated output format. A 100% score for this metric indicates that all the model’s outputs are interpretable and can be directly parsed for downstream object retrieval, ultimately enabling the construction of tabletop scenes.
>
> ## Generalization ability on full-room scene generation.
>
> We appreciate your valuable suggestion. First, it is important to note that generating reasonable 3D layouts from task instruction is not an inherent capability of LLMs. Our core contribution lies in endowing LLMs with 3D spatial reasoning abilities through fine-tuning on our well-constructed tabletop dataset that includes scenes with diverse 3D layouts. This process enables LLMs to generate plausible tabletop layouts from task instructions by learning the underlying spatial constraints and relational patterns specific to tabletop scenes.
>
> Regarding full-room layout generation, we recognize it as an interesting task. While full-room layouts and tabletop layouts share a similar format, they **differ fundamentally in critical aspects: object scales, object categories, and input task instruction**. As a result, LLMs fine-tuned exclusively on our tabletop dataset cannot be naively applied to full-room layout generation without further adaptation.
>
> Notably, if appropriate **task-layout paired data** for full-room scenes were available, our method could be similarly applied by fine-tuning on such data, thereby enabling the generation of reasonable full-room layouts. In future work, we plan to explore approaches for jointly generating full-room and tabletop scenes to produce plausible scenes for locomotion tasks.

---

> > ### Comment · Reviewer_2SRJ · 2025-08-08
> >
> > Thank you for the response. It has addressed my concerns. Although the generalization issue is not entirely resolved, I still consider this a solid piece of work and have decided to maintain my score.

---

### Official Review · Reviewer_MB7H · 2025-07-03

**Clarity:** 2
**Significance:** 3
**Originality:** 3
**Rating:** 5
**Confidence:** 4

**Summary:**

This paper introduces a novel task: task-oriented tabletop scene generation. To support this task, the authors curate a large-scale dataset, MesaTask-10K, containing 10,700 tabletop scenes that are semi-automatically generated and refined through manual collection, accompanied by relevant annotations. The proposed framework, MesaTask, leverages LLMs with a chain-of-thought reasoning process, termed the Spatial Reasoning Chain, to generate tabletop scenes. Comprehensive experiments are conducted to evaluate both the scene generation capabilities of the framework and the utility of the proposed dataset.

**Questions:**

1. Can your method handle cluttered or messy scenes where objects are irregularly piled rather than neatly arranged?

2. Please also see weakness

**Ethical Concerns:**

["NO or VERY MINOR ethics concerns only"]

**Final Justification:**

For the robot manipulation experiment, the robotic affordance detection component makes sense to me. It demonstrates the potential of a data collection pipeline—such as detecting affordances, sampling grasp poses, planning motion to the grasp pose, and executing the grasp—to generate robot manipulation data. Other methods could also be applied.

I believe the proposed data generation pipeline and the resulting dataset will benefit research in both scene generation and robotic manipulation. Therefore, I am raising my score to accept.

**Limitations:**

Yes

**Quality:**

3

**Strengths And Weaknesses:**

Strengths

1. The paper presents a large-scale and high-quality dataset for tabletop scene generation, MesaTask-10K, which has the potential to benefit future work in this area.

2. First using generative and perceptual models to obtain a rough scene layout, followed by manual refinement of the layout for data collection is a creative approach.

3. The chain-of-thought mechanism for LLM and the associated training procedure is well-designed.

4. Extensive experiments are performed to validate the framework’s performance.

Weakness

1. The abstract states that the motivation of the work is to generate task-relevant scenes for robot manipulation; however, the paper does not include any robotics experiments to support this claim.

2. The scene graphs used rely on relatively simple spatial relationships, which limits the diversity and complexity of the generated scenes.

---

> ### Author Rebuttal · Authors · 2025-07-31
>
> Thank you for your recognition of our work and your constructive feedback. Below, we address your questions and the weaknesses:
> ## Robotics experiments concern.
>
> Our work focuses on generating diverse, simulation-ready tabletop scenes. The scenes we generate exhibit strong physical plausibility and contain a large number of articulated objects, which are sourced from PartNet Mobility[1]—a widely used dataset for manipulation tasks validated by the community. All these features collectively offer rich interactable tabletop scenes for manipulation tasks.
>
> To further evaluate the physical manipulability of scenes in the constructed dataset, we conduct a **robotic affordance detection** experiment using the ManipVQA[2] model. Specifically, we select 50 multiple complex scenes containing a mug and frame the task as: "detect the graspable region of the mug." The handle of the mug is annotated as the ground-truth bounding box, and the experimental result yields a bounding box average precision (AP) of 0.93 at an Intersection over Union (IoU) threshold of 0.5, and 0.67 at an IoU threshold of 0.75. This result is close to that in ManipVQA. This demonstrates that existing affordance detection tasks can be effectively applied to our scenes. This setup assesses the physical plausibility of scene design by measuring the model’s ability to accurately localize graspable regions.
>
> ## Scene graphs' simple spatial relationships limit scene diversity and complexity.
>
> In our method, the scene graph is first generated within the spatial reasoning chain, after which the layout (detailed size and position) of each object in the scene is inferred. As a structured "skeleton" for subsequent layout generation, the scene graph enables the construction of complex scenes through a coarse-to-fine pipeline. This approach aligns with prior scene generation methods [3, 4, 5], which also leverage scene graphs to facilitate the generation of diverse, complex scene layouts.
>
> ## Model's ability to generate Messy tabletop scenes.
>
> Our method can generate messy scenes by incorporating simple post-processing steps. In real life, cluttered scenes often evolve from neat ones. For example, a messy table after a banquet starts as a neatly arranged one. Drawing on this idea, we first generate a neat scene using our method, then perturb the positions and rotations of objects in the scene, and finally use physical simulation to avoid object overlaps and floating, thereby enabling the generation of messy scenes.
>
> [1] Xiang F, Qin Y, Mo K, et al. Sapien: A simulated part-based interactive environment (CVPR 2020).
>
> [2] Huang S, Ponomarenko I, Jiang Z, et al. Manipvqa: Injecting robotic affordance and physically grounded information into multi-modal large language models (IROS 2024).
>
> [3] Liu et al. Controllable 3D Outdoor Scene Generation via Scene Graphs. (ICCV 2025)
>
>
> [4] Lin et al. InstructScene: Instruction-Driven 3D Indoor Scene Synthesis with Semantic Graph Prior. (ICLR 2024)
>
>
> [5] Yang et al. HOLODECK: Language Guided Generation of 3D Embodied AI Environments. (CVPR 2024)

---

> > ### Comment · Reviewer_MB7H · 2025-08-06
> >
> > Thank you for the response. My concerns have been addressed. I believe the data generation pipeline and the dataset will benefit research in scene generation and robot manipulation. I will raise my score.

---

> ### Author Response · Authors · 2025-08-06
>
> Dear Reviewer,
>
> Thank you once again for reviewing our paper. We would greatly appreciate it if you could take a moment to review our feedback and newly added experiments, and let us know if any concerns remain.
>
> Best regards,
>
> The Authors

---

### Decision · Program_Chairs · 2025-09-17

**Decision:**

Accept (spotlight)

**Comment:**

The paper introduces MesaTask-10K, a novel large-scale dataset of synthetic tabletop scenes designed for training robots to interpret human instructions and perform manipulation tasks. The authors propose a new task: task-oriented tabletop scene generation. They address the challenge of bridging the gap between high-level task instructions and the generation of plausible 3D tabletop scenes by introducing a Spatial Reasoning Chain. This chain decomposes the generation process into object inference, spatial interrelation reasoning, and scene graph construction. They also present MesaTask, an LLM-based framework that utilizes this reasoning chain, enhanced with DPO algorithms, to generate physically plausible and task-aligned scenes.

Strengths:
- The MesaTask-10K dataset is considered a large-scale and high-quality resource that could significantly benefit future research in tabletop scene generation.
- The semi-automatic data generation approach is novel. The Spatial Reasoning Chain is well-designed.
- The method's modular design is logically coherent and amenable to integration with chain-of-thought style LLM-based architectures.
- The paper is well-written and easy to follow.

No major weaknesses after the discussion. The clarifications and additional experiments provided during the rebuttal period were crucial in solidifying the decision to recommend acceptance. The authors are highly encouraged to incorporate them into the camera-ready version.

Overall, the paper stands out due to its novel task formulation and the creation of a valuable, large-scale dataset for task-oriented tabletop scene generation. The Spatial Reasoning Chain offers a promising approach to bridging the gap between high-level instructions and 3D scene creation. While there were initial concerns about experimental validation and dataset details, the authors addressed these concerns adequately during the rebuttal period. The reviewers converge to an agreement that the proposed pipeline and dataset will benefit research in scene generation and robotic manipulation. The thorough responses from the authors, coupled with the potential impact of the dataset and method, justify recommending acceptance as a Spotlight paper.